# Untangling stability and gain modulation in cortical circuits with multiple interneuron classes

Hannah Bos[1†], Christoph Miehl[2,3†], Anne-Marie Michelle Oswald[2,3], Brent Doiron[1,2,3,4,5*]

[1]Department of Mathematics, University of Pittsburgh, Pittsburgh, United States; [2]Department of Neurobiology, University of Chicago, Chicago, United States; [3]Grossman Center for Quantitative Biology and Human Behavior, University of Chicago, Chicago, United States; [4]Department of Neuroscience, University of Pittsburgh, Pittsburgh, United States; [5]Department of Statistics, University of Chicago, Chicago, United States

*For correspondence:
bdoiron@uchicago.edu

†These authors contributed equally to this work

Competing interest: The authors declare that no competing interests exist.

## eLife Assessment

This paper explores how diverse forms of inhibition impact firing rates in models for cortical circuits. In particular, the paper studies how the network operating point affects the balance of direct inhibition from SOM inhibitory neurons to pyramidal cells, and disinhibition from SOM inhibitory input to PV inhibitory neurons. This is an **important** issue as these two inhibitory pathways have largely been studied in isolation. A combination of analytical calculations and direct numerical simulations provides **convincing** evidence that the interplay of these inhibitory circuits can separately control network gain and stability.

**Abstract** Synaptic inhibition is the mechanistic backbone of a suite of cortical functions, not the least of which are maintaining network stability and modulating neuronal gain. In cortical models with a single inhibitory neuron class, network stabilization and gain control work in opposition to one another – meaning high gain coincides with low stability and vice versa. It is now clear that cortical inhibition is diverse, with molecularly distinguished cell classes having distinct positions within the cortical circuit. We analyze circuit models with pyramidal neurons (E) as well as parvalbumin (PV) and somatostatin (SOM) expressing interneurons. We show how, in E – PV – SOM recurrently connected networks, SOM-mediated modulation can lead to simultaneous increases in neuronal gain and network stability. Our work exposes how the impact of a modulation mediated by SOM neurons depends critically on circuit connectivity and the network state.

## Introduction

While inhibition has been long measured (*Lloyd, 1946*; *Hartline et al., 1956*; *Eccles et al., 1954*), the past 22 years have witnessed a newfound appreciation of its diversity. The invention and widespread use of cell-specific labeling and optogenetic control (*Fenno et al., 2011*), combined with the detailed genetic and physiological characterization of cortical interneurons (*Markram et al., 2004*; *Jiang et al., 2015*) has painted a complex picture of a circuit. The standard cortical circuit now includes (at a minimum) SOM and PV expressing interneuron classes, with distinct synaptic interactions between these classes as well as with pyramidal neurons (*Pfeffer et al., 2013*; *Tremblay et al., 2016*; *Kepecs and Fishell, 2014*; *Jiang et al., 2015*; *Campagnola et al., 2022*). This additional complexity presents

some clear challenges (*Cardin, 2018*; *Wood et al., 2017*; *Ferguson and Cardin, 2020*; *Urban-Ciecko and Barth, 2016*; *Yavorska and Wehr, 2016*), foremost being to uncover how functions that were previously associated with inhibition in a broad sense should be distributed over diverse interneuron classes.

Inhibition has been long identified as a physiological or circuit basis for how cortical activity changes depending upon processing or cognitive needs (*Isaacson and Scanziani, 2011*). Inhibition has been implicated in the suppression of neuronal activity (*Adesnik et al., 2012*; *Kato et al., 2017*; *Haider et al., 2013*; *Adesnik, 2017*), gain control of pyramidal neuron firing rates (*Phillips and Hasenstaub, 2016*; *Katzner et al., 2011*; *Ferguson and Cardin, 2020*; *Silver, 2010*) and correlated neuronal fluctuations (*Okun and Lampl, 2008*), rhythmic population activity (*Atallah and Scanziani, 2009*; *Womelsdorf et al., 2014*), spike timing of pyramidal neurons (*Wehr and Zador, 2003*; *Berman and Maler, 1998*), and gating synaptic plasticity (*Paille et al., 2013*; *Wu et al., 2022*; *Canto-Bustos et al., 2022*). However, inhibition must also prevent runaway cortical activity that would otherwise lead to pathological activity (*Haider et al., 2013*; *Ozeki et al., 2009*; *Veit et al., 2017*), enforcing constraints on how inhibition can modulate pyramidal neuron activity. This broad functional diversity has prompted theorists to build circuit models to expose how the synaptic structure and dynamics of inhibition affect network behavior.

Cortical models with excitatory and inhibitory neurons have a long history of study (*Wilson and Cowan, 1972*; *Griffith, 1963*). Models with just a single inhibitory interneuron class have successfully explained a wide range of cortical behavior; from contrast-dependent nonlinearities in cortical response (*Rubin et al., 2015*; *Ozeki et al., 2009*), to the genesis of irregular and variable spike discharge (*van Vreeswijk and Sompolinsky, 1996*; *Brunel, 2000*), to the mechanisms underlying high-frequency cortical network rhythms (*Wang, 2010*; *Bos et al., 2016*). However, these models explore how inhibition supports a single function or network dynamic. In this way, these models are unique and are designed to capture only a restricted dataset. This is a reflection of the limitations imposed by considering only one type of inhibitory interneuron in a cortical circuit.

An attractive hypothesis is that distinct interneurons are within-class functionally homogeneous, yet each class performs functions that are distinct from those of the other classes (*Kepecs and Fishell, 2014*; *Hattori et al., 2017*; *Wang et al., 2004*). In recent years, computational studies have used circuit models with multiple inhibitory neuron types to study distinct roles of inhibitory neurons like effects on network oscillations (*Ter Wal and Tiesinga, 2021*; *Veit et al., 2023*), circuit modulation e.g., via locomotion or attention (*Dipoppa et al., 2018*; *Poort et al., 2022*; *Myers-Joseph et al., 2024*), network stabilization (*Garcia del Molino et al., 2017*; *Litwin-Kumar et al., 2016*; *Palmigiano et al., 2023*; *Kumar et al., 2023*), and many more (*Richter and Gjorgjieva, 2022*; *Waitzmann et al., 2024*; *Hertäg and Sprekeler, 2019*; *Keijser and Sprekeler, 2022*; *Aponte et al., 2021*; *Wilmes and Clopath, 2019*; *Sadeh et al., 2017*; *Pedrosa and Clopath, 2020*; *Edwards et al., 2024*). A prominent example of the division of labor hypothesis is that PV neurons are well-positioned to provide network stability (*Wang et al., 2004*), allowing SOM neurons to modulate the circuit.

We use previously developed multi-interneuron cortical circuit models (*Litwin-Kumar et al., 2016*; *Kuchibhotla et al., 2017*; *Garcia del Molino et al., 2017*; *Mahrach et al., 2020*; *Veit et al., 2023*; *Palmigiano et al., 2023*; *Waitzmann et al., 2024*; *Kumar et al., 2023*) with the goal of giving a mechanistic understanding of how modulations of SOM neurons affect various circuit components. At the core of our study, SOM modulations can impact excitatory neurons differentially through either a direct inhibitory path onto excitatory neurons or an indirect disinhibitory path via PV interneurons. Depending on the recurrent connections from excitatory or PV neurons onto SOM neurons these distinct SOM modulations can have, sometimes non-intuitive, influence on circuit firing rates, network stability, stimulus gain, and stimulus tuning. Our theoretical framework offers an attractive platform to probe how interneuron circuit structure determines gain and stability which may generalize well beyond the sensory cortices where these interneuron circuits are currently best characterized.

## Results
### The inhibitory and disinhibitory pathways of the E − PV − SOM circuit

There is strong in vivo evidence that SOM interneurons play a critical role in the modulation of cortical response (*Urban-Ciecko and Barth, 2016*; *Yavorska and Wehr, 2016*). However, the complex wiring

between excitatory and inhibitory neurons (*Tremblay et al., 2016*; *Pfeffer et al., 2013*; *Jiang et al., 2015*; *Campagnola et al., 2022*) presents a challenge when trying to expose the specific mechanisms by which SOM neurons modulate cortical response. Two distinct inhibitory circuit pathways are often considered when disentangling the impact of SOM inhibition on excitatory neuron (E) response: an inhibitory SOM → E pathway or a disinhibitory SOM → PV → E pathway.

Experimental studies find different, at first glance contradicting, effects of SOM neurons on E. In one line of study, SOM neuron activity seems to directly inhibit E neurons. Increased SOM activity resulted in decreased activity in E neurons in studies of layer 2/3 mouse visual cortex (*Adesnik et al., 2012*; *Adesnik, 2017*). Similarly, decreased SOM activity resulted in increased E neuron activity in the piriform cortex (*Canto-Bustos et al., 2022*), and other studies (*Wang and Yang, 2018*). In another line of study, changes in E activity following SOM perturbation seems to follow from disinhibitory pathways. For example, silencing layer 4 SOM neurons in mouse somatosensory cortex resulted in decreased activity of E neurons (*Xu et al., 2013*). Taken together, these two lines of studies seem in opposition to one another, with SOM neuron activity either suppressing or increasing E activity. This response dichotomy prompted us to consider what physiological and circuit properties of the E – PV

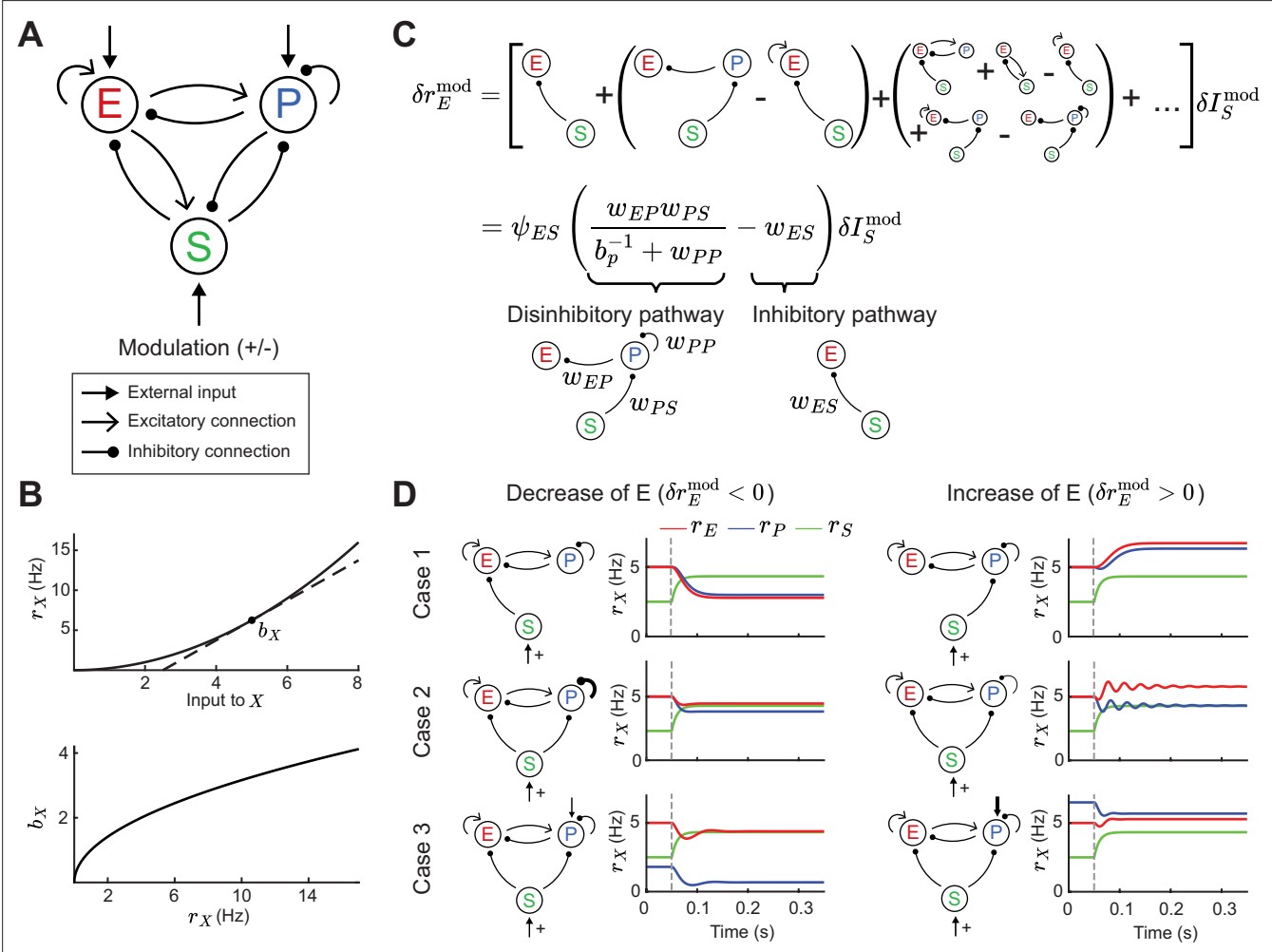

**Figure 1.** Tradeoff between two inhibitory motifs in the excitatory (E) – parvalbumin (PV) – somatostatin (SOM) cortical circuit. (**A**) Sketch of the full E – PV – SOM network model. A positive or negative modulatory input is applied to the SOM neurons. (**B**) Transfer function (top) and population gain $b_X$ (bottom) for neuron population $X = \{E, P, S\}$ (see *Equation 4*). (**C**) Top: Relation between modulation of input to the SOM population $\delta I_S^{\mathrm{mod}}$ and changes in E rates $\delta r_E^{\mathrm{mod}}$ when summing over all possible paths (see *Equation 10*). Bottom: After summing over all paths. Sketches visualize the tradeoff between the inhibitory and disinhibitory pathways (see *Equation 15*). (**D**) Positive SOM modulation at 0.05 s (gray dashed line) decrease (left, $\delta r_E^{\mathrm{mod}} < 0$) or increase (right, $\delta r_E^{\mathrm{mod}} > 0$) the E rate $r_E$ (red line). Case 1: Add connection of SOM → E or SOM → PV population. Case 2: Change strength of self-inhibition of PV population. Case 3: Change the rate of PV neurons.

– SOM circuit are critical determinants of whether an increase in SOM neuron activity results in an increase or a decrease in E neuron response.

An answer to this question requires consideration of the full recurrent connectivity within the E – PV – SOM neuron circuit, as opposed to analysis restricted to just the SOM → E and SOM → PV → E sub motifs within the circuit. We set up a recurrent network where we model the firing rates of E, PV, and SOM neurons (*Figure 1A*; see Methods), as has been done by similar studies of the E – PV – SOM cortical circuit (*Litwin-Kumar et al., 2016*; *Kuchibhotla et al., 2017*; *Garcia del Molino et al., 2017*; *Mahrach et al., 2020*; *Veit et al., 2023*; *Palmigiano et al., 2023*; *Waitzmann et al., 2024*; *Kumar et al., 2023*). The key factors differentiating PV and SOM neurons in our model are that PV neurons inhibit other PV neurons, while SOM neurons do not, and that PV neurons receive external (sensory) input while SOM neurons receive modulatory input. Using our model we ask how a modulation of the SOM neuron activity (via $\delta I_S^{\mathrm{mod}}$) results in a modulation of E neuron activity ($\delta r_E^{\mathrm{mod}}$). Examples of such modulation includes suppressed vasoactive intestinal-peptide (VIP) inhibition onto SOM neurons (*Pi et al., 2013*), activation of pyramidal cells located outside the circuit yet preferentially projecting to SOM neurons (*Adesnik et al., 2012*), and direct cholinergic modulation of SOM neurons (*Kuchibhotla et al., 2017*; *Urban-Ciecko and Barth, 2016*).

The model response is nonlinear, with neurons in each population having an expansive nonlinear transfer function (*Figure 1B*; top, see *Equation 4*), consistent with many experimental reports (*Priebe and Ferster, 2008*; *Romero-Sosa et al., 2021*). To understand which circuit parameters can influence the sign of E rate changes, we apply a widely used concept: if the modulation of SOM inputs ($\delta I_S^{\mathrm{mod}}$) is sufficiently small we can linearize around a given dynamical state of the model. At the neuronal level, this linearization defines a cellular gain $b_X$ ($X \in \{E, P, S\}$) from the transfer function (*Figure 1B*; bottom). At the network level the linearization involves the entire circuit (*Garcia del Molino et al., 2017*; *Litwin-Kumar et al., 2016*; *Palmigiano et al., 2023*) and yields:

$$\delta r_E^{\mathrm{mod}} = L_{ES}\delta I_S^{\mathrm{mod}}, \tag{1}$$

where $L_{E,S}$ is the transfer coefficient between SOM and E neuron modulations. In principle, $L_{E,S}$ depends on the synaptic weight matrix $\mathbf{W}$ in which each element $w_{X,Y}$ defines the coupling between neuron classes (with $X, Y = \{E, P, S\}$), as well as the cellular gain $b_X$ of all neuron classes (see Methods). In principle, $L_{E,S}$ depends on twelve parameters: the nine synaptic couplings within the E – PV – SOM circuit and the three cellular gains. This large parameter space convolutes any analysis of modulations; our study provides a framework to navigate this complexity.

To begin, it is instructive to express the effect of SOM on E based on all possible synaptic pathways. Intuitively, the effect of SOM modulation on E rates can be understood by an infinite sum of synaptic pathways with increasing order of synaptic connections (*Figure 1C*; top). Hence, the changes in SOM rate affect E rates via the monosynaptic pathway SOM → E, disynaptic pathways SOM → PV and PV → E or SOM → E and E → E, trisynaptic pathways, etc. Fortunately, the sum can be simplified so that just two network motifs determine the sign of changes in E rates (*Figure 1C*; bottom, see *Equation 15*). These motifs reflect both the disinhibitory component of the network (the SOM → PV → E and PV → PV connections) and the inhibitory component (SOM → E connections). Whether the full motif is biased towards the inhibitory or disinhibitory pathway depends on the connection strengths $w_{EP}$, $w_{PS}$, $w_{ES}$, and $w_{PP}$. Furthermore, since the PV gain depends on the operating point of the network, the tradeoff between the two pathways can be controlled by changes in PV rates. In particular, since PV gain increases with PV rates (*Figure 1B*), then $L_{ES}$ can transition from effectively inhibitory for low PV activity (small $b_P$) to effectively disinhibitory for higher PV activity (large $b_P$). We remark that other connections and the activity of the E and SOM neurons only contribute to the amplitude but not the sign of the effective pathway. This is because these other components are part of the prefactor $\psi_{ES}$, which is always positive in the case of a stable circuit (see Methods, *Equation 15*).

Therefore, for a certain choice of connectivity and input parameters, SOM modulation yields a decrease in E rates ($\delta r_E^{\mathrm{mod}} < 0$), as reported from neuronal recordings in layers 2 and 3 of visual cortex of mice (*Adesnik et al., 2012*; *Adesnik, 2017*; *Figure 1D*; left). A different choice of parameters yields an increase in E rates ($\delta r_E^{\mathrm{mod}} > 0$), consistent with recordings from layer 4 neurons from the somatosensory cortex of mice (*Xu et al., 2013*; *Figure 1D*; right). Our analysis of how synaptic pathways determine the sign of $\delta r_E^{\mathrm{mod}}$ (*Figure 1C*) provides a framework to discuss the possible mechanistic reasons for this discrepancy. Specifically, this change in E rate for the same SOM modulation can in principle

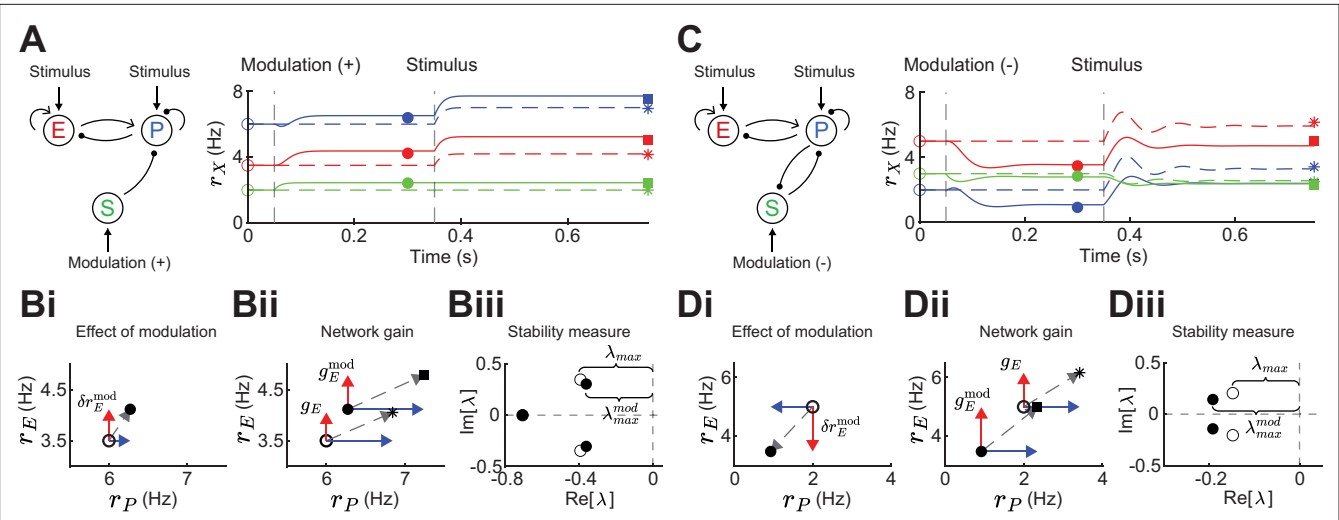

**Figure 2.** Gain and stability in excitatory (E) – parvalbumin (PV) – somatostatin (SOM) circuits. (**A**) Left: Sketch of a disinhibitory network with stimulus input onto E and PV populations and positive SOM modulation. Right: Numerical E (red), PV (blue), and SOM (green) rate dynamics of the case with positive SOM modulation at 0.05 s (solid line), and the case without modulation (dashed line). Stimulus presentation at 0.35 s. Symbols indicate calculated values based on *Equation 1* and *Equation 2*. (**B**) Measures to quantify the effect of SOM modulation: (**i**) Effect of modulation on E ($\delta r_E^{\mathrm{mod}}$) and PV rates, (**ii**) calculation of network gain with ($g_E^{\mathrm{mod}}$) and without ($g_E$) SOM modulation, $\Delta g = g_E^{\mathrm{mod}} - g_E = 0.12$, (**iii**) calculation of stability measure with ($\lambda_{\max}^{\mathrm{mod}}$) and without ($\lambda_{\max}$) SOM modulation, $\Delta \lambda = \lambda_{\max} - \lambda_{\max}^{\mathrm{mod}} = -0.03$. (**C**) Same as A for a negative SOM modulation in a disinhibitory circuit with feedback PV → SOM. (**D**) Same as B for a negative SOM modulation with (**ii**) $\Delta g = 0.35$, and (**iii**) $\Delta \lambda = 0.04$ (only maximum eigenvalues shown).

follow from differences in: direct inhibition of E via SOM versus disinhibition of E via SOM (*Figure 1D*; Case 1), strong versus weak self-inhibition of PV (*Figure 1D*; Case 2), or low versus high firing rates of PV (*Figure 1D*; Case 3). Hence, differential modulations in E rate response might follow from any of those circuit or cellular factors.

In sum, while the full E – PV – SOM recurrent circuit invokes a multitude of polysynaptic pathways, a tradeoff between the inhibitory and disinhibitory pathway does indeed determine the modulatory influence of SOM neurons upon E neuron activity. Having now identified the central role of these two pathways, in the following sections we investigate how they control network stability and the stimulus – response gain of E neurons.

## Gain modulation and stability measures

In the following, we ask how SOM modulation can affect stimulus representation. In most primary sensory cortices, sensory stimulus information arrives at E and PV neurons via feedforward connections (*Tremblay et al., 2016*). Therefore, we model stimulus as a feedforward input onto E and PV populations (*Figure 2A*; left). An important feature of cortical computation is gain modulation, which refers to changes in the sensitivity of neuron activity to changes in a driving input (*Silver, 2010*; *Ferguson and Cardin, 2020*; *Williford and Maunsell, 2006*). Many experimental studies suggest that inhibitory neurons play an important role in gain modulation (*Ferguson and Cardin, 2020*; *Isaacson and Scanziani, 2011*). In the following, we analyze how a modulation via SOM neurons can affect the stimulus – response gain of the E population.

To motivate our analysis we compare the influence of a stimulus with and without SOM modulation in a disinhibitory pathway (*Figure 2A*). Since the linearization framework outlined above allows us to calculate the effect of a SOM modulation on E rates (*Figure 2Bi*; $\delta r_E^{\mathrm{mod}}$), we can further ask how a SOM modulation affects the gain of the network. We define the network gain as the rate change of the E population in response to a change in the stimulus ($\delta \mathbf{I}^{\mathrm{stim}}$), assuming that stimuli target E and PV populations.

$$
\begin{aligned}
g_E &= L_{EE}\delta I_E^{\mathrm{stim}} + L_{EP}\delta I_P^{\mathrm{stim}} \\
&= \psi_g\left(\left((b_P^{-1} + w_{PP}) - b_S w_{PS} w_{SP}\right)\delta I_E^{\mathrm{stim}} - \left(w_{EP} - b_S w_{ES} w_{SP}\right)\delta I_P^{\mathrm{stim}}\right).
\end{aligned}
$$

(2)

Here, network gain measures the sensitivity of E rates owing to the activity of the full recurrent circuit in response to a change in input. This is opposed to the cellular gain $b_E$ which measures the sensitivity of E rates to a change in the input current to E neurons (*Figure 1B*; top). The expression in *Equation 2* allows us to calculate the difference in network gain $\Delta g = g_E^{\text{mod}} - g_E$ with and without SOM modulation when a stimulus is presented (*Figure 2A and Bii*). Since the cellular gains $b_E$ and $b_P$ depend upon the operating point about which the circuit dynamics are linearized, the tradeoff between amplification and cancellation can be controlled through an external modulation (e.g. via SOM) that shifts this point.

In addition to network gain, we will also measure how SOM modulation affects the stability of the network. Unstable firing rate dynamics are typified by runaway activity when recurrent excitation is not stabilized by recurrent inhibition (*Ozeki et al., 2009*; *van Vreeswijk and Sompolinsky, 1996*; *Wilson and Cowan, 1972*; *Griffith, 1963*). Stability in a dynamical system is quantified by the real parts of the eigenvalues of the Jacobian matrix. If the real parts of all eigenvalues are less than zero, the system is stable. To quantify stability, we measure the distance of the largest real eigenvalue (i.e. least negative) to zero (*Figure 2Biii*; Methods). To compare stability for the modulated versus the unmodulated case, we subtract the largest real eigenvalues $\Delta\lambda = \lambda_{\text{max}} - \lambda_{\text{max}}^{\text{mod}}$. Therefore, if $\Delta\lambda > 0$ stability increases via SOM modulation, and if $\Delta\lambda < 0$ stability decreases. In the example of purely disinhibitory influence of SOM modulation, network gain is increased (*Figure 2Bii*; $\Delta g = 0.12$) and stability slightly decreases (*Figure 2Biii*; $\Delta\lambda = -0.03$). Hence, in this network example increase in network gain is accompanied by decreases in network stability. By contrast, in a network with feedback PV → SOM neurons (*Figure 2C*), a negative modulation of SOM neurons leads to decreases in E and PV rates (*Figure 2Di*) while increasing both, network gain (*Figure 2Dii*; $\Delta g = 0.35$) and stability (*Figure 2Diii*; $\Delta\lambda = 0.04$).

Therefore, the direction and magnitude of gain and stability changes depend on the connectivity details of the inhibitory circuit. In the following sections, we dissect how firing rates and synaptic weights within the E – PV – SOM circuit contribute to modulations of network gain and stability.

## Gain and stability controlled by feedforward SOM inhibition

We start by considering a network without connections between the E – PV network and the SOM population (*Figure 3i*). To compare network gain across different network states we consider a grid of possible firing rates ($r_E, r_P$). A given network state is found by determining the external input required to position the network at that rate (see Methods). For each E – PV rate pair, we linearize the network dynamics (i.e. determine the cellular gains $b_X$) and compute the network gain via *Equation 2* (*Figure 3ii*). It is immediately apparent that network gain is largest for high E rates and low PV rates. Gain modulation is most effective when it connects two network states that are orthogonal to a line of constant gain (*Figure 3ii*; gray lines). Thus, for most network states the highest gain increase occurs for modulations that increase E neuron rates while simultaneously decreasing PV neuron rates. In a similar fashion, we consider how stability depends on network activity ($r_E, r_P$) (*Figure 3iii*). Network dynamics are most stable for large PV and low E neuron rates. Discontinuities in the lines of constant stability

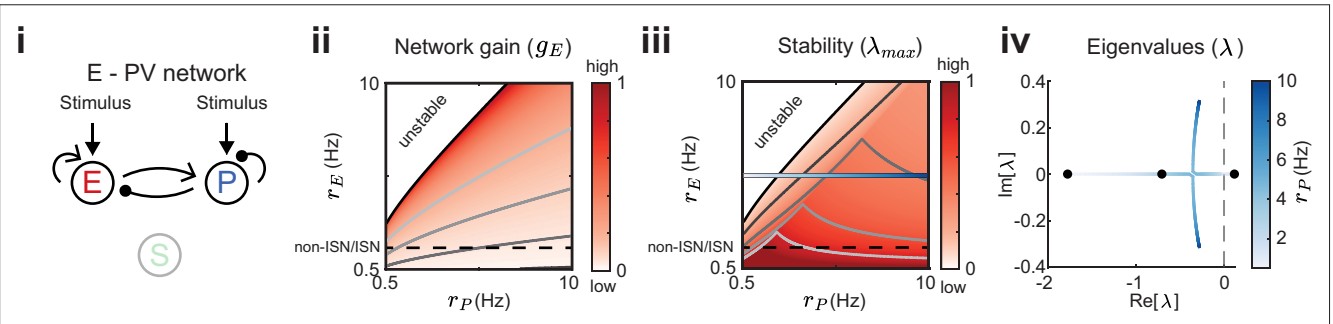

**Figure 3.** Network gain and stability in the excitatory (E) – parvalbumin (PV) network. Network sketch (**i**), firing rate grid ($r_E, r_P$) in the form of a heatmap for normalized network gain $g_E$ (**ii**) and normalized stability $\lambda_{\text{max}}$ (**iii**), and the eigenvalues for changing PV rates $r_P$ (**iv**) for a network without connections between the E – PV network and somatostatin (SOM). Every value in the heatmap is a fixed point of the population rate dynamics. The color denotes normalized network gain (*Equation 2*) or normalized stability (*Figure 2Biii*). Lines of constant network gain and stability are shown in gray (from dark to light gray in steps of 0.2). The black line marks where the rate dynamics become unstable. The black dashed line separates inhibition stabilized network (ISN) from non-ISN regime. Blue line in iii indicates the parameters for which the eigenvalues are shown in (**iv**).

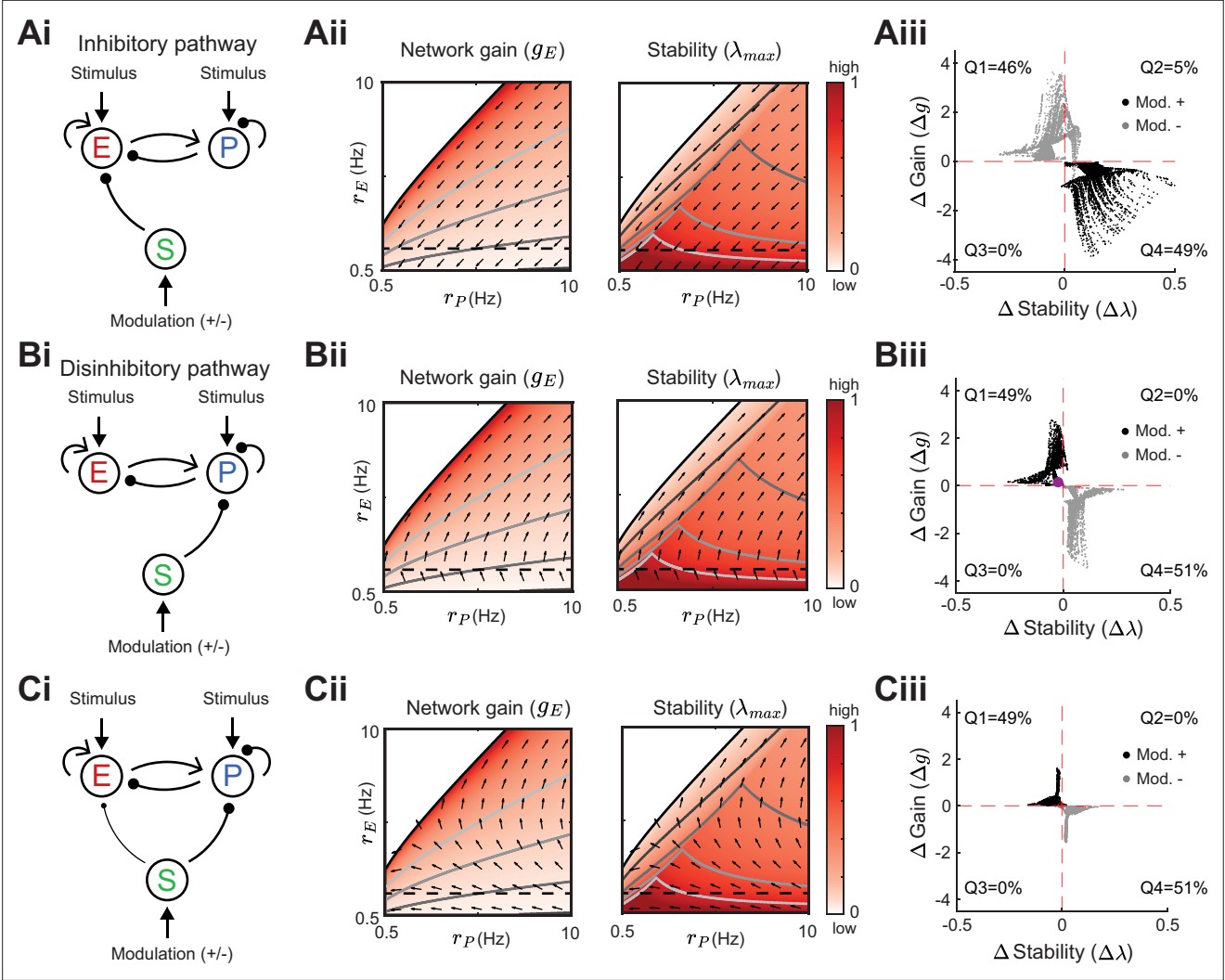

**Figure 4.** Modulation of somatostatin (SOM) neurons with feedforward SOM connectivity. (**A**) Network sketch (**i**), firing rate grid $(r_E, r_P)$ in the form of a heatmap for normalized network gain and stability (**ii**), and modulation measures $\Delta$ Gain ($\Delta g$) and $\Delta$ Stability ($\Delta \lambda$) (**iii**), for a network with SOM → E connection (inhibitory pathway). The arrows indicate in which direction a fixed point of the rate dynamics is changed by a positive SOM modulation. All arrow lengths are set to the same value. The modulation measure quantifies the change in stability and gain from an initial condition in the $(r_E, r_P)$ grid for a positive (black dots) and negative (gray dots) SOM modulation. Q1-Q4 indicates the percent of data points in the respective quadrant (only $|\Delta g| > 0.1$ and $|\Delta \lambda| > 0.01$ are considered). (**B**) Same as A for a network with SOM → PV connection (disinhibitory pathway). The purple dot in Biii is the case of *Figure 2A*. (**C**) Same as A for a network with SOM → E and SOM → PV connections. SOM rate $r_S = 2$ Hz in all panels.

follow from discontinuities in the dependence of eigenvalues on PV rate (*Figure 3iv*; see Methods). In total, we have an inverse relationship between these two network features, where high gain is accompanied by low stability and vice-versa (compare heatmaps *Figure 3ii and iii*). This 'tangling' of gain and stability places a constraint on network modulations, ultimately limiting the possibility of high gain responses.

We next expand our network and include SOM neurons in order to consider how their modulation can affect network gain and stability. For now, we neglect feedback from E or PV populations onto SOM. Consequently, SOM neuron modulation can only affect the stability and gain of E neurons by changing the dynamic state of the E – PV subcircuit. Positive or negative input modulations to SOM neurons increase or decrease their steady-state firing rate, which in turn affects the steady-state rates of the E and PV neurons. To build intuition we first consider only the SOM → E connection and set the SOM → PV connection to zero, thereby isolating the inhibitory pathway (*Figure 4Ai*). A specific modulation can be visualized as a vector $(\Delta r_E, \Delta r_P)$ in the firing rate grid (*Figure 4Aii*). The direction of the vector indicates where the E – PV network state would move to if SOM neurons are weakly

positively modulated. We remark that the modulation ($\Delta r_E, \Delta r_P$) not only depends on the feedforward SOM projections to E and PV neurons, but also on the dynamical regime (i.e. linearization) of the unmodulated state ($r_E, r_P$). Applying a positive modulation to SOM neurons causes the E and PV rates to decrease (*Figure 4Aii*; arrows). We quantify the effect of all the possible modulations in the ($r_E, r_P$) grid on network gain and stability by calculating the difference in network gain ($\Delta g$) and stability ($\Delta \lambda$) before and after SOM modulation. For almost all cases, network gain and stability have an inverse relationship to each other. For a positive SOM modulation, network gain decreases while stability increases (*Figure 4Aiii*; black dots in the $\Delta \lambda > 0$ and $\Delta g < 0$ quadrant). Similarly, for a negative SOM modulation, network gain mostly increases while stability decreases (*Figure 4Aiii*; gray dots in the $\Delta \lambda < 0$ and $\Delta g > 0$ quadrant).

We next consider only the SOM → PV connection and set SOM → E to zero, isolating the disinhibitory pathway (*Figure 4Bi*). If the unmodulated network state has low E rates then the modulation vector field shows a transition from decreases in PV rates to increases in PV rates. A network response where PV rates increase with a decrease in the inputs to PV population is often labeled a paradoxical effect (*Ozeki et al., 2009*; *Tsodyks et al., 1998*; *Litwin-Kumar et al., 2016*). Therefore, with a disinhibitory pathway we can get changes from non-paradoxical to paradoxical responses when switching from non-inhibition stabilized network (non-ISN) to an inhibition stabilized network (ISN) (*Litwin-Kumar et al., 2016*; *Ozeki et al., 2009*; *Tsodyks et al., 1997*), indicated by $\Delta r_P < 0$ for low $r_E$ yet shifting to $\Delta r_P > 0$ for larger $r_E$ (*Figure 4Bii*). Similar to the inhibitory pathway, network gain and stability are inversely related (*Figure 4Biii*). If we extend our analysis by including weak SOM → E connectivity (*Figure 4Ci*), the SOM → PV connection continues to dominate and maintains a mostly disinhibitory effect on E neurons for high rates (*Figure 4Cii*). The vector field changes so that the modulation now strongly increases gain but also shifts the circuit more directly into the unstable region while keeping the inverse relationship between gain and stability changes (*Figure 4Ciii*).

In sum, our analysis shows that modulation of the E − PV circuit via feedforward SOM modulation results in an inverse relationship between network gain and stability. Hence, an increase in gain is accompanied by a decrease in stability and vice versa. Intuitively, the reason why the inverse relationship follows for inhibitory and disinhibitory pathways (and their mixture) is that the firing rate grid ($r_E, r_P$ heatmap) does not depend on how the SOM neurons inhibit the E − PV circuit. Different SOM inputs only modify the direction of rate changes following SOM modulation (arrows). Since the underlying firing rate grid already has an inverse relationship, then any modulation of SOM neurons will in turn have an inverse relationship between network gain and stability. These results prompt the question: can a cortical circuit be modulated through inhibition to a higher gain regime without compromising network stability? In the next section, as indicated by the motivating example (*Figure 2C*), we show how feedback to SOM neurons can shift the E − PV − SOM circuit from a low to a high gain state while maintaining stability.

## Recurrent inputs to SOM neurons allow modulations to increase both gain and stability

Neglecting feedback connections to SOM in the E − PV − SOM circuit makes SOM activity simply an intermediate step in a feedforward modulation of the E − PV subcircuit. In this section, we consider how the E → SOM and PV → SOM interactions determine how an external modulation to SOM neurons affects E network gain and stability.

We first remark that by adding feedback E connections onto SOM neurons, changes in SOM rates can now affect the underlying heatmaps in the ($r_E, r_P$) grid, meaning that high or low regions of network gain and stability in the space of E and PV rates depend on SOM connectivity and rates. This is because *Equation 2* has a dependency on the SOM rates ($r_S$) through the cellular gain ($b_S$). In the case of an inhibitory pathway with feedback from E → SOM, SOM modulations can change gain and stability in the same direction (*Figure 5*). Dependent on the initial rates in the ($r_E, r_P$) grid, a positive SOM modulation can lead to an increase in both, network gain and stability (*Figure 5Aiii,Biii,Ciii*). The higher the SOM rates, the more likely it becomes for a positive modulation to result in a gain and stability increase. However, we note that the network gain changes with the highest amplitude are accompanied by decreases in stability. Similarly, in the example of a disinhibitory pathway with feedback from PV → SOM, SOM modulation can lead to changes of network gain and stability in the same direction (*Figure 5—figure supplement 1*). Here a negative SOM modulation can lead to increases

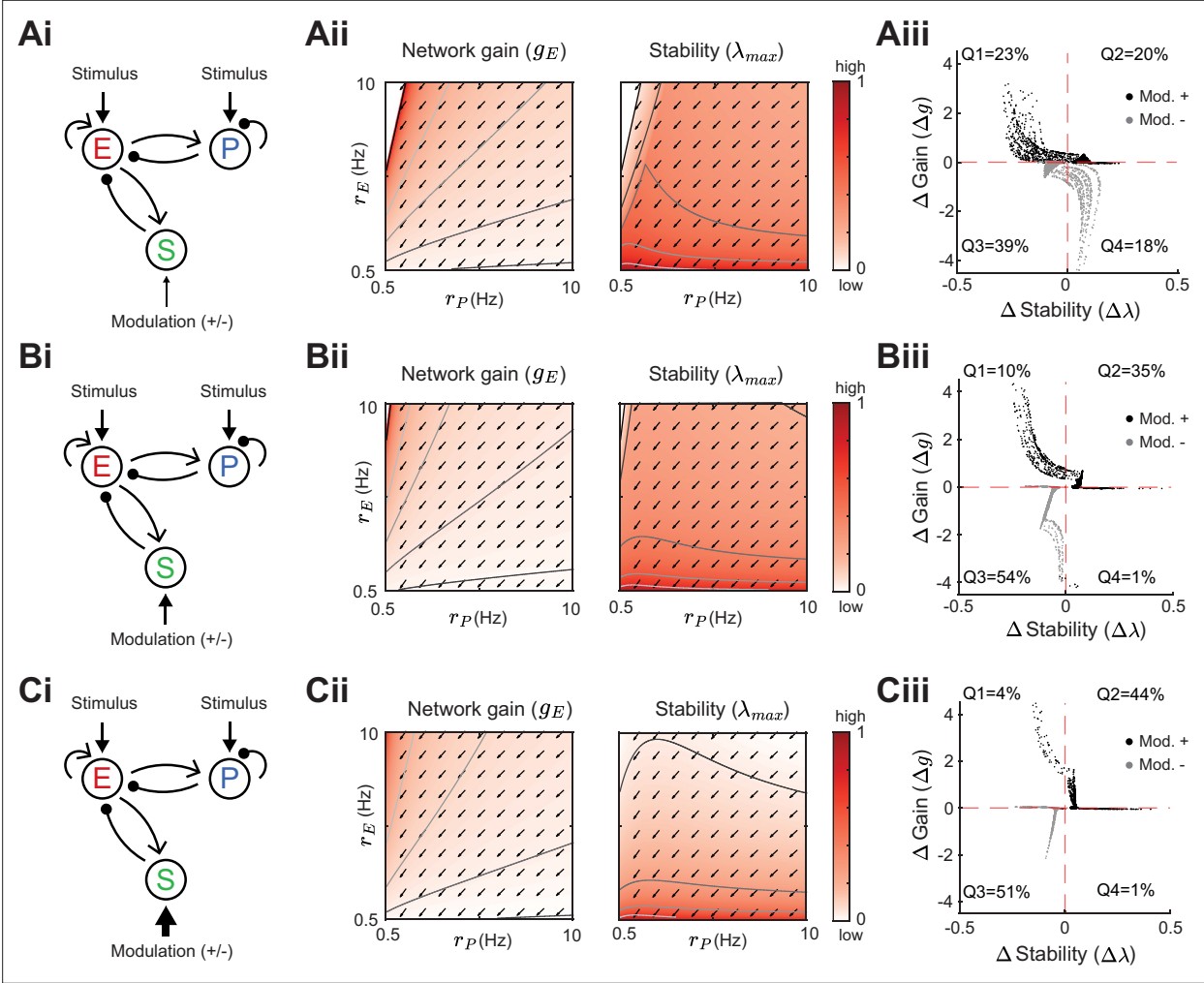

**Figure 5.** Modulation of somatostatin (SOM) neurons with excitatory (E) to SOM feedback. Heatmaps and modulation measures as defined in *Figures 3 and 4* for a network with an inhibitory pathway and E → SOM feedback. Left to right: Network sketch (**i**), normalized network gain ($g_E$) and stability ($\lambda_{max}$) (**ii**), and modulation measures Δ Gain ($\Delta g$) and Δ Stability ($\Delta \lambda$) (**iii**). Top to bottom: increase of the SOM firing rate from $r_S = 1$ Hz (**A**), to $r_S = 2$ Hz (**B**), $r_S = 3$ Hz (**C**). The arrows indicate in which direction a fixed point of the rate dynamics is changed by a positive SOM modulation.

The online version of this article includes the following figure supplement(s) for figure 5:

**Figure supplement 1.** Modulation of somatostatin (SOM) neurons with parvalbumin (PV) to SOM feedback.

**Figure supplement 2.** Percent of data points in Q1-Q4 when changing somatostatin (SOM) firing rate.

in both, network gain and stability. Furthermore, we confirm that for both E to SOM feedback and PV to SOM feedback these results are robust for a large range of SOM firing rates (*Figure 5—figure supplement 2*).

In summary, adding a recurrent connection onto SOM neurons from the E (*Figure 5*) or PV (*Figure 5—figure supplement 1*) neurons allows network gain and stability to change in the same direction for a SOM modulation. This follows since recurrent connections affect the underlying rate grid (heatmaps). Here, a SOM modulation can shift the network state across the lines of constant network gain and stability in a way that increases both, network gain and stability. This 'disentangling' of the inverse relation between gain and stability allows SOM-mediated modulations to sample a broader range of responses.

## Gain and stability in stochastically forced E – PV – SOM circuits

To confirm that our results do not depend on our approach of a linearization around a fixed point, we numerically simulate similar networks as shown above (*Figure 2*) in which the E and PV population

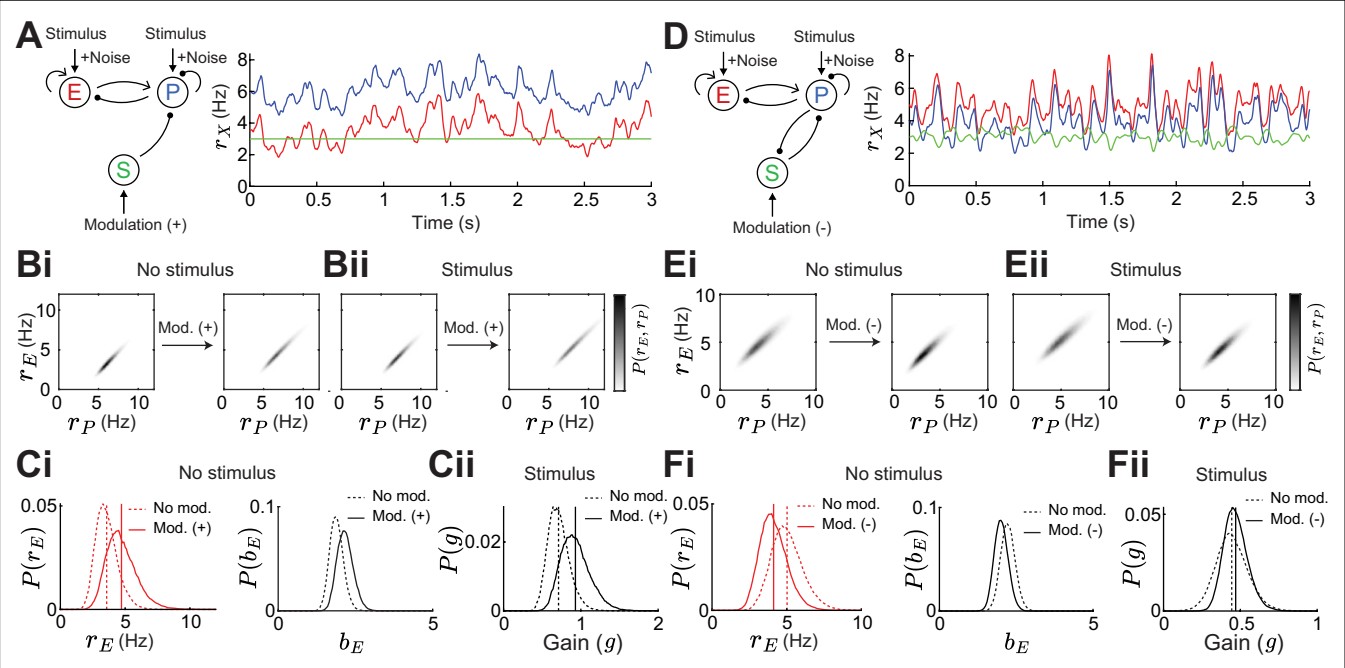

**Figure 6.** Gain and stability in noisy excitatory (E) – parvalbumin (PV) – somatostatin (SOM) circuits. (**A**) Left: Sketch of a disinhibitory network with stimulus plus noise input onto E and PV populations and positive SOM modulation. Right: Numerical E (red), PV (blue), and SOM (green) rate dynamics. (**B**) Distribution of E and PV rates for a positive SOM modulation without (**i**) and with a stimulus (**ii**) (**C**) Changes in the distribution of E rates $r_E$ (i, left), E population gain $b_E$ (i, right) and network gain $g$ (ii) with vs without SOM modulation. Variance of E rates for no SOM modulation is 0.7 and with SOM modulation 1.3. (**D**) Same as A for a negative SOM modulation in a disinhibitory circuit with feedback PV → SOM. (**E**) Same as B for negative SOM modulation. Variance of E rates for no SOM modulation is 1.1 and with SOM modulation 0.8. (**F**) Same as C for negative SOM modulation.

receive slow varying, large amplitude noise (*Figure 6A*). This leads to noisy rate dynamics sampling a large subspace of the full firing rate grid ($r_E, r_P$) and thus any linearization would fail to describe the network response. In this stochastically forced network we explore how adding an SOM modulation or a stimulus affects this subspace (*Figure 6B*). To quantify stability without linearization, we assume that a network is more stable the lower the mean and variance of E rates. This is because very stable networks can better quench input fluctuations (*Kanashiro et al., 2017*; *Hennequin et al., 2018*). To quantify gain, we calculate the change in E rates when adding the stimulus, yet having identical noise realizations for stimulated and non-stimulated networks (Methods).

For the disinhibitory network without feedback a positive SOM modulation decreases stability due to increases in the mean and variance of E rates (*Figure 6Ci*) while the network gain increases (*Figure 6Cii*). As seen before (*Figure 2A and B*), stability and gain change in opposite directions in a disinhibitory circuit without feedback. Adding feedback PV → SOM and applying a negative SOM modulation increases both, stability and gain and, therefore, disentangles the inverse relation also in a noisy circuit (*Figure 6D–F*). This gives numerical support that our results do not depend on the assumption of linearization.

## Influence of weight strength on network gain versus stability

In the previous sections, we have studied how the population firing rates influence network gain and stability in various network configurations through changes in the cellular gain and inhibitory versus disinhibitory pathways with and without feedback to SOM. However, following from our motivating example, the decrease or increase of E rates to SOM modulation can depend on the exact strength of certain synaptic weights (*Figure 1D*; Case 2). In this section, we show in detail how changes in synaptic weight strength can affect network gain and stability. We consider four cases: a network with a biased inhibitory pathway ($w_{ES} > w_{PS}$) (*Figure 7Ai-Aiv*), or a biased disinhibitory pathway ($w_{ES} < w_{PS}$) (*Figure 7Bi-Biv,*) and we distinguish between the network being in the non-ISN regime where the E → E connection ($w_{EE}$) is weak (*Figure 7*) and the ISN regime with strong $w_{EE}$ (*Figure 7—figure supplement 1*). We note that throughout we keep the rates of all populations fixed (see Methods).

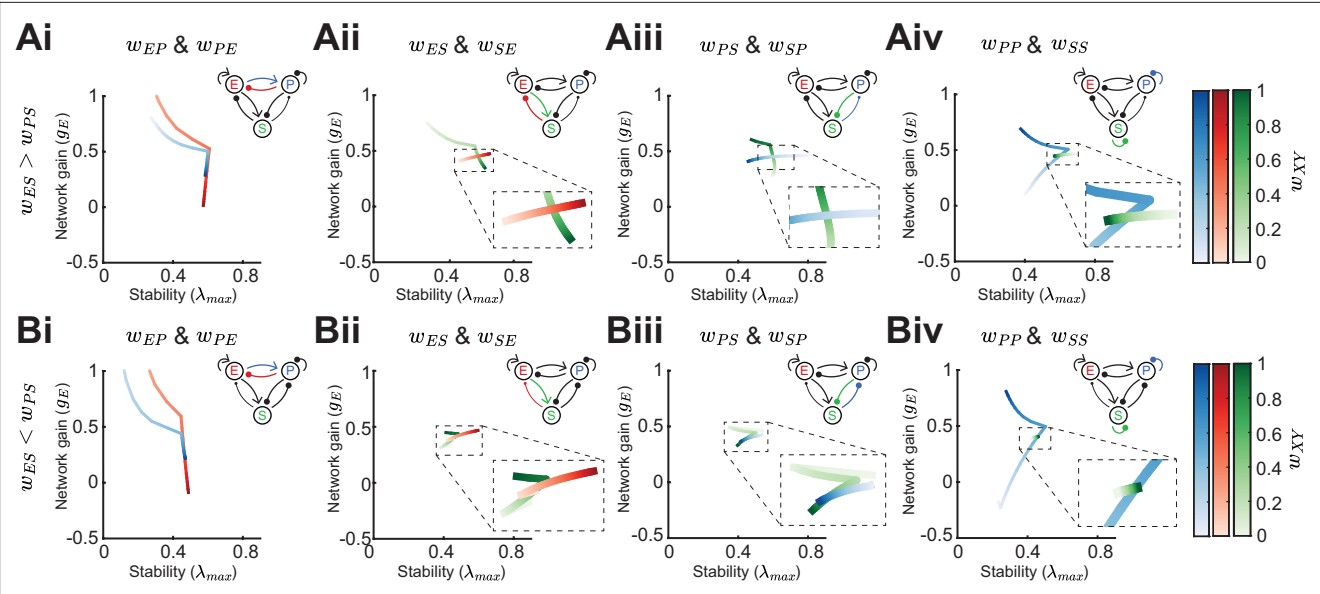

**Figure 7.** Effect of synaptic weight strength on network gain and stability. (**A**) Effect of synaptic weight change on network gain ($g_E$) and stability ($\lambda_{max}$) in a network biased to inhibitory somatostatin (SOM) influence ($w_{ES} > w_{PS}$). We change the strength of one weight at a time, either $w_{EP}$ or $w_{PE}$ (**i**), $w_{ES}$ or $w_{SE}$ (**ii**), $w_{PS}$ or $w_{SP}$ (**iii**), or $w_{PP}$ or $w_{SS}$ (**iv**). Colorbar indicates the weight strength, red corresponds to weights onto excitatory neurons (**E**) blue onto parvalbumin (PV), and green onto SOM. (**B**) Same as A but in a network biased to disinhibitory SOM influence ($w_{ES} < w_{PS}$). The networks are in the non-inhibition stabilized network (ISN) regime ($w_{EE}$ is weak) and all the rates are fixed $r_E = 3$, $r_P = 5$, $r_S = 0.5$. Dashed rectangles represent zoom-in.

The online version of this article includes the following figure supplement(s) for figure 7:

**Figure supplement 1.** Effect of synaptic weight strength on network gain and stability (inhibition stabilized network, ISN regime).

For weakening either the connection from PV → E ($w_{EP}$) or E → PV ($w_{PE}$) the network gain drastically increases and is mostly accompanied by decrease in stability (**Figure 7Ai, Bi**). However, if the influence of SOM on E is biased to be inhibitory, increases in network gain can lead to slight increases in stability (**Figure 7Ai**; strong $w_{EP}$ or $w_{PE}$). This follows from the discontinuity of the stability measure, as we have already pointed out in a previous section (**Figure 3iv**; see Methods). The influence of the feedback connection E → SOM ($w_{SE}$) depends on the bias of SOM connectivity. For inhibitory biased networks, increasing the strength of $w_{SE}$ reduces gain (**Figure 7Aii**), while for disinhibitory biased networks it leads to an increase of gain (**Figure 7Bii**). The connection SOM → E ($w_{ES}$) moderately increases both, stability and gain (**Figure 7Aii, Bii**). Similarly, the influence of the feedback connection PV → SOM ($w_{SP}$) is opposed for the inhibitory biased versus disinhibitory biased case and the SOM → PV connection ($w_{PS}$) changes gain and stability in the same direction (**Figure 7Aiii, Biii**).

An important distinction between PV and SOM neurons is that PV neurons are strongly connected to other PV neurons, while SOM → SOM ($w_{SS}$) coupling has not been found in the mouse sensory neocortex (**Pfeffer et al., 2013**; **Tremblay et al., 2016**; **Urban-Ciecko and Barth, 2016**; **Campagnola et al., 2022**). The PV self coupling strength can have a large effect on both network gain and stability (**Figure 7Aiv,Biv**). An interesting aspect of PV → PV ($w_{PP}$) coupling is that it appears that there is an optimal weight strength for maximal stability. On the other hand, SOM self coupling has only minimal effect on gain and stability.

In summary, changing synaptic weights have often non-intuitive effects on network gain and stability. Network gain always either decreases or increases when changing the strength of a single weight, but the direction in which network gain changes depends on inhibitory biased versus disinhibitory biased, e.g., as shown for changing $w_{SE}$ (**Figure 7Aii, Bii**). This can be understood from **Equation 2**, which directly shows how the direction (sign) of network gain changes depends on the respective weight parameter. For stability, discontinuities appear making the direction of change for stability dependent on the absolute weight strengths of the respective weight, e.g., increasing PV self connection strength first increases stability while when further increasing the weight strength leads to a decrease of stability (**Figure 7Aiv, Biv**). In contrast to network gain, it is difficult to gain

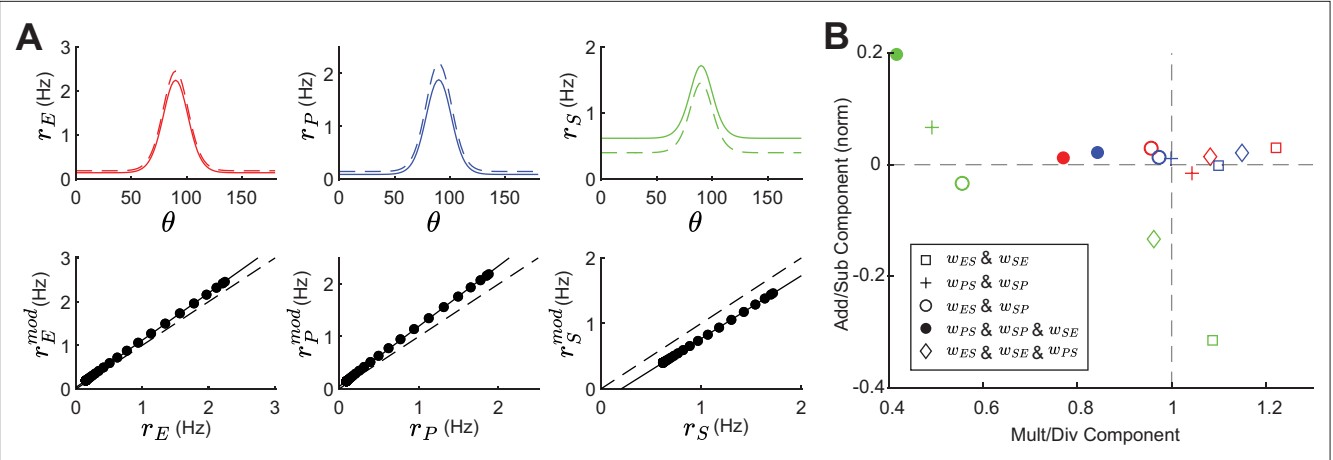

**Figure 8.** Tuning curve changes induced by somatostatin (SOM) modulation depend on network connectivity. (**A**) Top: Tuning curves of excitatory (E) (red), parvalbumin (PV) (blue), and SOM (green) populations in a network with connections SOM → E and SOM → PV and a feedback connection E → SOM ($w_{ES}, w_{PS}, w_{SE} \neq 0$). Solid lines represent the tuning curve before modulation and dashed lines after a negative SOM modulation. Bottom: Linear regression of unmodulated versus modulated rates (black dots: unmodulated versus modulated rate pairs, gray solid line: fit, gray dashed line: unity line). (**B**) Multiplicative/divisive component versus additive/subtractive component for different network connectivities. Add/sub component is normalized to the maximum rate response. Diamond case is shown in panel A.

intuition about the dependence of stability on the weights because the eigenvalues have a complex relationship to all the weights and the maximum eigenvalue might show nonlinear dynamics (as shown in *Figure 3iv*).

## Modulation of SOM neurons can have diverse effects on tuning curves

In the previous sections, we measured network gain as the increase of E neuron activity in response to a small increase in stimulus intensity. We now extend our analysis to E – PV – SOM circuits with distributed responses, whereby individual neurons are tuned to a particular value of a stimulus (i.e. the preferred orientation of a bar in a visual scene or the frequency of an acoustic tone). In what follows the stimulus $\theta$ is parametrized with an angle ranging from 0°–180°.

We begin by giving the E and PV populations feedforward input which is tuned to $\theta = 90°$ with a Gaussian profile (see *Equation 19*). Providing tuned input leads to a tuned response at E, PV and SOM populations (*Figure 8A*; top, solid lines). Even though the SOM population does not receive tuned external input, the tuning of SOM is expected since they receive input from tuned E. A small negative modulation of the SOM population can modify the tuning properties of all populations (*Figure 8A*; top, dashed lines). In experimental studies that optogenetically activate or inactivate inhibitory populations, changes in tuning curves are often characterized as a linear transformation containing shifting (additive or subtractive) and scaling (multiplicative or divisive) components (*Phillips and Hasenstaub, 2016*; *Arandia-Romero et al., 2016*). By fitting a line to the rates before versus after SOM modulation we can quantify the respective components (*Figure 8A*; bottom). The slope of the fitted line corresponds to the magnitude of the multiplicative (slope >1) or divisive (slope <1) component while the intercept with the y-axis reveals the additive (intersect >0) or subtractive (intersect <0) component of tuning curve changes. In the example of a network with connections from SOM → E and SOM → PV and a feedback connection from E → SOM (as shown in *Figure 8A*), modulation of SOM leads to subtractive and divisive changes at SOM and additive and multiplicative changes at E and PV populations (*Figure 8B*; diamond).

For other network configurations, changes in tuning following a negative SOM modulation can be based on different components. For example, in a network with SOM → E, SOM → PV connections and PV → SOM feedback all populations have an additive and divisive component (*Figure 8B*; filled circles).

In sum, tuning curve changes following from SOM modulation depend on the underlying network configuration and can differ largely in their components.

## Discussion

Cortical inhibition is quite diverse, with molecularly distinguished cell classes having distinct placement within the cortical circuit (*Markram et al., 2004*; *Tremblay et al., 2016*; *Pfeffer et al., 2013*; *Jiang et al., 2015*; *Campagnola et al., 2022*). Cell-specific optogenetic perturbations are a critical probe used to relate circuit wiring to cortical function. In many cases, a preliminary analysis of these new optogenetic datasets involves building circuit intuition only from the dominant direct synaptic pathways while neglecting indirect or disynaptic pathways. This is understandable given the complexity of the circuit; however, this is precisely the situation where a more formal modeling approach can be very fruitful. Toward this end, recent modeling efforts both at the large (*Billeh et al., 2020*; *Markram et al., 2015*) and smaller (*Litwin-Kumar et al., 2016*; *Kuchibhotla et al., 2017*; *Mahrach et al., 2020*; *Garcia del Molino et al., 2017*; *Veit et al., 2023*; *Ter Wal and Tiesinga, 2021*; *Palmigiano et al., 2023*; *Hertäg and Sprekeler, 2019*; *Keijser and Sprekeler, 2022*; *Richter and Gjorgjieva, 2022*; *Waitzmann et al., 2024*; *Kumar et al., 2023*; *Aponte et al., 2021*; *Edwards et al., 2024*) scales have incorporated key aspects of interneuron diversity. These studies typically explore which aspects of cellular or circuit diversity are required to replicate a specific experimental finding.

In our study, we provide a general theoretical framework that dissects the full E – PV – SOM circuit into interacting sub-circuits. We then identify how specific inhibitory connections support both network stability and E neuron gain control; two ubiquitous functions often associated with inhibition (*Ozeki et al., 2009*; *Haider et al., 2013*; *Ferguson and Cardin, 2020*; *Isaacson and Scanziani, 2011*). In this way, our approach gives an expanded view of the mechanics of cortical function when compared to more classical results that focus only on how circuit structure supports a single feature of cortical dynamics. The theoretical framework we develop can be adopted to investigate other structure-function relationships in complicated multi-class cortical circuits, like thalamocortical loops, cortical layer-specific connectivities, or circuits including also VIP neurons.

### Division of labor between PV and SOM interneurons

Compelling theories for both network stability (*Ozeki et al., 2009*; *van Vreeswijk and Sompolinsky, 1996*; *Griffith, 1963*) and gain control *Sutherland et al., 2009*; *Stern et al., 2018* have been developed using simple cortical models having only one inhibitory neuron class. Thus, network stability and gain control do not necessarily require cortical circuits with diverse inhibition. What our study points out is that SOM neurons are ideal for modulating firing rate changes, network gains, and stability.

Two key circuit features support our division of labor breakdown. First, E neurons and PV neurons experience very similar types of inputs. Both receive excitatory drive from upstream areas (*Tremblay et al., 2016*), and both receive strong recurrent excitation, as well as PV- and SOM-mediated inhibition (*Pfeffer et al., 2013*; *Campagnola et al., 2022*). This symmetry in the synaptic input to E and PV neurons allows PV neurons to dynamically track E neuron activity. Consequently, any spurious increase in excitatory drive to E neurons, that could cause a cascade of E population activity due to recurrent E → E connections, is quickly countered by an associated increase in PV inhibition. Second, SOM neurons do not connect to other SOM neurons (*Pfeffer et al., 2013*; *Urban-Ciecko et al., 2015*; *Jiang et al., 2015*; *Campagnola et al., 2022*). SOM neurons do provide strong inhibition to E neurons, and this lack of input symmetry makes them less fit to stabilize E neuron activity than PV neurons. However, it is precisely the lack of SOM neuron self-inhibition that allows a high gain for any top-down modulatory signal to induce a change in E neuron response. A large component of the analysis in our manuscript is devoted to establishing this circuit-based view of a division of inhibitory labor in E – PV – SOM cortical circuits. However, there is also evidence for the reverse labor assignment, namely that optogenetic perturbation of PV neurons can shift E neuron response gain (*Wilson et al., 2012*; *Atallah et al., 2012*; *Seybold et al., 2015*), and SOM neurons can suppress E neuron firing which in principle would also quench runaway E neuron activity (*Adesnik et al., 2012*; *Adesnik, 2017*).

In our study, both PV and SOM neurons affect stimulus – response gain and stability. We show that the PV firing rate strongly modulates both gain and stability, often in opposing directions (*Figure 4*). Similarly, changing the connection strength of the E – PV subcircuit has the largest effect on network gain (*Figure 7*). That said, SOM neurons can control how E and PV neurons interact. A key result of our study is that feedforward SOM inhibition of the E – PV circuit leads to an inverse relationship between network gain and stability. Increases (decreases) in gain are often followed by decreases (increases) in stability (*Figure 4*). However, adding recurrent feedback onto SOM neurons can disentangle this

inverse relationship. Indeed, for many circuit parameter choices gain and stability can increase or decrease together (*Figure 5*). This suggests that feedback onto SOM neurons is an important feature to have more flexibility for circuit computation.

An interesting observation is that network gain depends on firing rates of E, PV, and SOM neurons at the moment of stimulus presentation (*Figure 3ii*; *Figure 4Aii, Bii, Cii*; *Figure 5Aii,Bii, Cii*). Hence any change in input to the circuit can affect the response gain to a stimulus presentation, in line with experimental evidence which suggests that changes in inhibitory firing rates and changes in the behavioral state of the animal lead to gain modifications (*Ferguson and Cardin, 2020*).

There are circuit and cellular distinctions between PV and SOM neurons that were not considered in our study, but could nonetheless still contribute to a division of labor between network stability and modulation. Pyramidal neurons have widespread dendritic arborizations, while by comparison PV neurons have restricted dendritic trees (*Markram et al., 2004*). Thus, the dendritic filtering of synaptic inputs that target distal E neurons dendrites would be quite distinct from that of the same inputs onto PV neurons. PV neurons target both the cell bodies and proximal dendrites of both PV and E neurons (*Markram et al., 2004*; *Tremblay et al., 2016*; *Di Cristo et al., 2004*), so that the symmetry of PV inhibition onto PV and E neurons as viewed by action potential initiation is maintained. In stark contrast, SOM neurons inhibit the distal dendrites of E neurons (*Markram et al., 2004*). Dendritic inhibition has been shown to gate burst responses in pyramdial neurons greatly reducing cellular gain (*Larkum et al., 2004*; *Mehaffey et al., 2005*), and theoretical work shows how such gating allows for a richer, multiplexed spike train code (*Naud and Sprekeler, 2018*; *Keijser and Sprekeler, 2022*; *Hertäg and Sprekeler, 2019*). Furthermore, dendritic inhibition is localized near the synaptic site for E → E coupling, and modeling (*Yang et al., 2016*) and experimental (*Adler et al., 2019*) work shows how such dendritic inhibition can control E synapse plasticity. This implies that SOM neurons may be an important modulator not only of cortical response but also of learning.

The E − PV − SOM cortical circuit is best characterized in superficial layers of sensory neocortex (*Pfeffer et al., 2013*; *Tremblay et al., 2016*; *Urban-Ciecko and Barth, 2016*). However, cell densities and connectivity patterns of interneuron populations change across the brain (*Kim et al., 2017*) and across cortical layers (*Tremblay et al., 2016*; *Jiang et al., 2015*). Our circuit-based division of labor thus predicts that any differences in inhibitory connectivity compared to the one we studied will be reflected in changes of the roles that interneurons play in distinct cortical functions.

## Influence of synaptic strength in the E − PV − SOM circuit

In most of our studies, the distinction between different circuits is based on the existence or non-existence of a synaptic connection. For example, the distinction between inhibitory and disinhibitory circuits can be made by setting the other connection to zero (*Figure 4A and B*). However, the exact synaptic strength of a connection relative to the strength of all other connection strengths in the circuit is an important determinant of circuit response. Small changes can switch the sign of how SOM modulation affects rates (*Figure 1C and D*) or change the stability and network gain of the circuit (*Figure 7*). Hence, our analysis suggests that including short- or long-term plasticity dynamics of synaptic weight strength can have profound impacts on the circuit.

Short-term synaptic dynamics in cortical circuits often show net depression (*Zucker and Regehr, 2002*), however, the E → SOM connection facilitates with increasing pre-synaptic activity (*Tremblay et al., 2016*; *Reyes et al., 1998*; *Thomson, 1997*; *Yavorska and Wehr, 2016*; *Beierlein et al., 2003*; *Urban-Ciecko and Barth, 2016*). Indeed, prolonged activation of E neurons recruits SOM activity through this facilitation (*Beierlein et al., 2003*). Thus, this enhanced gain control would require a strong and long-lasting drive to E neurons to facilitate the E → SOM synapses. Recent computational work has shown how distinct short-term plasticity dynamics at inhibitory synapses impact auditory processing (*Park and Geffen, 2020*; *Seay et al., 2020*; *Phillips et al., 2017*), multiplexing (*Hertäg and Sprekeler, 2019*; *Naud and Sprekeler, 2018*; *Keijser and Sprekeler, 2022*), and SOM response reversal (*Waitzmann et al., 2024*).

Recent experimental work also finds subtype-specific long-term plasticity dynamics (*Lagzi et al., 2021*; *Udakis et al., 2020*; *Wu et al., 2022*). A prominent role of inhibition, and specifically SOM neurons, is the gating of synaptic plasticity at excitatory neurons (*Canto-Bustos et al., 2022*; *Miehl and Gjorgjieva, 2022*). Our work suggests that there are weight strengths for which the stability of

the circuit becomes maximal (*Figure 7*), therefore, a potential goal of long-term synaptic plasticity might be to keep the synaptic weight strength of inhibitory connections at an optimal value.

## Impact of SOM neuron modulation on tuning curves

Neuronal gain control has a long history of investigation (*Salinas and Thier, 2000*; *Ferguson and Cardin, 2020*; *Williford and Maunsell, 2006*), with mechanisms that are both bottom-up (*Schwartz and Simoncelli, 2001*) and top-down (*Reynolds and Heeger, 2009*; *Ruff et al., 2018*) mediated. A vast majority of early studies focused on single neuron mechanisms; examples include the role of spike frequency adaptation (*Ermentrout, 1998*), interactions between fluctuating synaptic conductances and spike generation mechanics (*Chance et al., 2002*; *Ly and Doiron, 2009*), and dendritic-dependent burst responses (*Larkum et al., 2004*; *Mehaffey et al., 2005*). These studies often dichotomized gain modulations into a simple arithmetic where they are classified as either additive (subtractive) or multiplicative (divisive) (*Silver, 2010*; *Williford and Maunsell, 2006*). More recently, this arithmetic has been used to dissect the modulations imposed by SOM and PV neuron activity onto E neuron tuning (*Lee et al., 2014*; *Atallah et al., 2012*; *Wilson et al., 2012*). Initially, the studies framed a debate about how subtractive and divisive gain control should be assigned to PV and SOM neuron activation. However, a pair of studies in the auditory cortex gave a sobering account whereby activation and inactivation of PV and SOM neurons had both additive/subtractive and multiplicative/divisive effects on tuning curves (*Phillips and Hasenstaub, 2016*; *Seybold et al., 2015*), challenging the tidy assignment of modulation arithmetic into interneuron class. Specifically, optogenetically decreasing SOM activity leads to mostly additive and multiplicative tuning curve changes in the mouse primary auditory cortex (*Phillips and Hasenstaub, 2016*), which in our model follows from strong E to SOM feedback.

Past modeling efforts have specifically considered how tuned or untuned SOM and PV projections combine with nonlinear E neuron spike responses to produce subtractive or divisive gain changes (*Seybold et al., 2015*; *Litwin-Kumar et al., 2016*). However, the insights in these studies were primarily restricted to feedforward SOM and PV projections to E neurons, and ignored E neuron recurrence within the circuit. We show that additive/subtractive and multiplicative/divisive changes in tuning properties can strongly depend on the underlying circuit connectivity, in line with large heterogeneity of subtractive and divisive gain control reported in various studies (*Seybold et al., 2015*; *Wilson et al., 2012*; *Lee et al., 2014*; *Atallah et al., 2012*; *Natan et al., 2017*).

## Limitations and future directions

Our study is based on a linearization approach, which only allows us to investigate the circuit dynamics close to a stable network state. While this makes our results mathematically tractable and more intuitive and we confirm that our results hold in the case with noisy inputs (*Figure 6*), an interesting future direction is to test if the results hold also in oscillatory or chaotic dynamical regimes.

Our model is based on two different inhibitory neuron populations, PV and SOM. Often inhibitory neurons are subdivided into (at a minimum) three populations PV, SOM, and VIP (*Pfeffer et al., 2013*). While we did not model VIP neurons explicitly, one possible source of SOM modulation is via VIP neurons. VIP neurons strongly connect to SOM cells, forming a disinhibitory pathway (*Pi et al., 2013*; *Pfeffer et al., 2013*). A possible extension of our model is to include VIP cells in the circuit, as has been done in previous studies (*Garcia del Molino et al., 2017*; *Palmigiano et al., 2023*; *Waitzmann et al., 2024*).

We note that it would be useful to apply our framework with a focus on a specific brain region and add all relevant cell types (at a minimum E, PV, SOM, and VIP) plus a dendritic compartment, in order to formulate much more precise experimental predictions. For example, a recent experimental study shows how optogenetic activation of SOM (and VIP) cells affect responses of pyramidal neurons in mouse primary auditory cortex to auditory stimuli (*Tobin et al., 2025*).

Furthermore, we study changes in tuning curves by assuming that the E and PV populations are tuned to a single orientation. A possible extension of our model is to study a ring attractor model with PV and SOM inhibitory neurons (*Rubin et al., 2015*), or study the tuning curve heterogeneity in balanced networks (*Hansel and van Vreeswijk, 2012*).

# Methods

## Population model

The population rate dynamics ($r_X$) of E, PV, and SOM neurons are described by a firing rate model (*Wilson and Cowan, 1972*)

$$\tau_X \frac{dr_X}{dt} = -r_X + f_X(q_X). \tag{3}$$

with $\tau_X$ being the rate time constant ($\tau_X = 10\text{ms}$ for all populations). The input to the circuit component $X$ is the linearly rectified sum over all presynaptic components $Y$ of synaptic weights $w_{XY}$ multiplied by the respective rate dynamics $r_Y$ plus external input $I_X : q_X = [\sum_Y (-1)^q w_{XY} r_Y + I_X]_+$. Here $X, Y$ either represent the excitatory (E), PV (P), or SOM (S) population with the exponent q=1 (q=2) if population $Y$ is inhibitory (excitatory). The nonlinear transfer functions are described by a power law.

$$f_X(q_X) = \alpha q_X^{\beta}. \tag{4}$$

To simplify our analysis we chose the same parameters $\alpha = 1/4$ and $\beta = 2$ for all populations (*Figure 1B*). We note that by choosing a linear transfer function ($\beta = 1$) the corresponding population gain term is constant for all inputs $b_X = \alpha$, and therefore there is no dependence of the gain and stability on the neuron firing rates.

In vector notation, *Equation 3* can be written as,

$$\mathbf{T}\frac{d\mathbf{r}}{dt} = -\mathbf{r} + \mathbf{f(q)} = -\mathbf{r} + \mathbf{f(Wr + I)}. \tag{5}$$

**Table 1.** Weight parameters.

| Figure | $w_{EE}$ | $w_{EP}$ | $w_{PE}$ | $w_{PP}$ | $w_{ES}$ | $w_{PS}$ | $w_{SE}$ | $w_{SP}$ |
|---|---|---|---|---|---|---|---|---|
| *Figure 1C*, Case 1 (left) | | 0.5 | | 0.6 | 0.2 | 0 | | |
| *Figure 1C*, Case 1 (right) | | | | | 0 | 0.2 | | |
| *Figure 1C*, Case 2 (left) | | | | 1 | | | | 0 |
| *Figure 1C*, Case 2 (right) | | 1 | | 0.1 | 0.5 | 0.6 | | |
| *Figure 1C*, Case 3 | | | | | | | | |
| *Figure 2A and B* | | | | | | 0.8 | 0 | |
| *Figure 2C and D* | | | | | 0 | | | 0.2 |
| *Figure 3* | | | | | | 0 | | |
| *Figure 4A* | | | | | 0.8 | | | |
| *Figure 4B* | | | | | 0 | 0.8 | | 0 |
| *Figure 4C* | 0.8 | | 1 | | 0.3 | | | |
| *Figure 5, Figure 5—figure supplement 2A* | | | | | 0.8 | 0 | 0.2 | |
| *Figure 5—figure supplement 1* | | | | 0.6 | | | | 0.2 |
| *Figure 5—figure supplement 2B* | | 0.5 | | | 0 | 0.8 | 0 | |
| *Figure 6A–C* | | | | | | 0.8 | | 0 |
| *Figure 6D–F* | | | | | | | | 0.2 |
| *Figure 8A and B,◊* | | | | | 0.5 | | 0.5 | 0 |
| *Figure 8B,□* | | | | | 0.8 | 0 | | |
| *Figure 8B,+* | | | | | 0 | 0.8 | 0 | |
| *Figure 8B,○* | | | | | 0.8 | 0 | | 0.5 |
| *Figure 8B,●* | | | | | 0 | 0.8 | 0.5 | |

with $\mathbf{T}$ being a diagonal matrix of rate time constants $\tau_X$, $\mathbf{r}$ the vector of firing rates $r_X$, $\mathbf{I}$ the vector of external inputs $I_X$, and $\mathbf{W}$ the synaptic connectivity matrix.

$$\mathbf{W} = \begin{pmatrix} w_{EE} & -w_{EP} & -w_{ES} \\ w_{PE} & -w_{PP} & -w_{PS} \\ w_{SE} & -w_{SP} & -w_{SS} \end{pmatrix}. \tag{6}$$

Note that in this notation we dropped the linear rectifier and assume only positive $\mathbf{q}$.

We summarize the weight parameters for each Figure in *Table 1*. Self-connection of SOM cells ($w_{SS}$) is always zero, besides in *Figure 7Aiv, Biv*. In *Figure 7*, we keep the strength of each weight at $w_{XY} = 0.5$ while changing the strength of only one weight (for the inhibitory case in *Figure 7A* we set $w_{PS} = 0.1$ and for the disinhibitory case we set $w_{ES} = 0.1$). In *Figure 7—figure supplement 1* we use the same parameters, besides the E → E weights are higher ($w_{EE} = 0.8$).

To generate the panels containing the grid of possible firing rates ($r_E, r_P$) we choose the external inputs to each population $I_X$ accordingly. The numerical results in *Figures 1D, 2A and C*, *Figures 6 and 8* are obtained via Euler integration with a timestep of 0.01.

## Calculation of modulation and gain

In the steady-state, the population rates are given by the self-consistent equation.

$$\mathbf{r} = \mathbf{f}(\mathbf{Wr} + \mathbf{I}). \tag{7}$$

Changes in the steady-state rates induced by small changes in the external rate $\mathbf{I}$ are given by *Litwin-Kumar et al., 2016*; *Garcia del Molino et al., 2017*.

$$\delta\mathbf{r} = \frac{d\mathbf{r}}{d\mathbf{I}}\delta\mathbf{I}. \tag{8}$$

The matrix $\mathbf{L} = \frac{d\mathbf{r}}{d\mathbf{I}}$ has been termed a response matrix and can be written as (*Garcia del Molino et al., 2017*),

$$\mathbf{L} = \left(\mathbf{B}^{-1} - \mathbf{W}\right)^{-1} = \left(\mathbf{1} - \mathbf{BW}\right)^{-1}\mathbf{B}. \tag{9}$$

Here 1 denotes the identity matrix, and $\mathbf{B}$ is defined as the diagonal matrix of cellular gains at the linearization points $b_X = \frac{df_X}{dq_X}(q_X^{ss})$ with $q_X^{ss}$ being the steady state input to the circuit component $X$. If all eigenvalues of $\mathbf{BW}$ are smaller than 1 the response matrix can be written as,

$$\mathbf{L} = \sum_{i=0}^{\infty} \left(\mathbf{BW}\right)^i \mathbf{B}. \tag{10}$$

The response of the E population $\delta r_E^{\text{mod}}$ to modulations of SOM $\delta I_S^{\text{mod}}$ following *Equation 10* can be expressed as,

$$\delta r_E^{\text{mod}} = \frac{dr_E}{dI_S^{\text{mod}}}\delta I_S^{\text{mod}} = b_S \sum_{i=0}^{\infty} \left(\mathbf{BW}\right)_{13}^i \delta I_S^{\text{mod}} \tag{11}$$

$$= b_S \Big( b_E w_{ES} - b_E^2 w_{EE} w_{ES} + b_E b_P w_{EP} w_{PS} \tag{12}$$

$$+ (b_P w_{EP} w_{PE} + b_S w_{ES} w_{SE} - b_E w_{EE}^2)b_E^2 w_{ES} \tag{13}$$

$$+ (b_E w_{EE} w_{EP} - b_P w_{EP} w_{PP})b_E b_P w_{PS} + ... \Big) \delta I_S^{\text{mod}} \tag{14}$$

Here $(\mathbf{BW})_{13}^i$ denotes the element in the first row and third column of the matrix. Our expression shows that the response matrix describes the summed effect of all possible pathways through the network whereby an externally applied signal could influence population E rates, as shown in *Figure 1D* (top).

Similarly, assuming that modulation only targets SOM neurons $\delta\mathbf{I} = (0, 0, \delta I_S^{\text{mod}})$, the rate change of excitatory neurons induced by modulation following *Equation 9* is given by

$$\delta r_E^{\text{mod}} = L_{ES}\delta I_S^{\text{mod}} = \frac{b_E^{-1}(b_P^{-1} + w_{PP})}{\det(\mathbf{B}^{-1} - \mathbf{W})}\left(\frac{w_{EP}w_{PS}}{b_P^{-1} + w_{PP}} - w_{ES}\right)\delta I_S^{\text{mod}}. \tag{15}$$

With $\psi_{ES} = b_E^{-1}(b_P^{-1} + w_{PP})/\det(\mathbf{B}^{-1} - \mathbf{W})$ being the prefactor in *Figure 1D*. If the system is stable, $\psi_{ES}$ is positive.

Network gain is defined as the rate change of neurons in response to a stimulus, assuming that stimuli target E and PV neurons $\delta\mathbf{I}^{\text{stim}} = (\delta I_E^{\text{stim}}, \delta I_P^{\text{stim}}, 0)$. The E neuron network gain is given by

$$g_E = \frac{dr_E}{d\mathbf{I}^{\text{stim}}}\delta\mathbf{I}^{\text{stim}} = L_{EE}\delta I_E^{\text{stim}} + L_{EP}\delta I_P^{\text{stim}}$$

$$= \psi_g\left(\left((b_P^{-1} + w_{PP}) - b_S w_{PS}w_{SP}\right)\delta I_E^{\text{stim}} - \left(w_{EP} - b_S w_{ES}w_{SP}\right)\delta I_P^{\text{stim}}\right). \tag{16}$$

This is the expression in *Equation 2* with prefactor $\psi_g = b_S^{-1}/\det(\mathbf{B}^{-1} - \mathbf{W})$. Again, for a stable system $\psi_g > 0$.

## Paradoxical responses and gain maximum

The response of PV to SOM modulation is given by

$$L_{PS} = \frac{(w_{EE} - b_E^{-1})w_{PS} - w_{ES}w_{PE}}{\det(\mathbf{B}^{-1} - \mathbf{W})}. \tag{17}$$

When SOM neurons only project to PV but not E neurons ($w_{ES} = 0$), the rate of PV neurons decreases for positive SOM modulation if the E – PV circuit is in the non-ISN regime ($w_{EE} < b_E^{-1}$) and increases otherwise (*Figure 4Bii*). The latter case has been termed paradoxical response (*Tsodyks et al., 1997*). If SOM neurons also project to E neurons, PV neurons get additional negative drive from the lack of E feedback yielding decreased PV rates even in the ISN regime (*Figure 4Aii, Cii*). Hence we only expect paradoxical responses if the product of connection strength $w_{ES}w_{PE}$ is small. Thus the observation of paradoxical responses of PV neurons in response to suppression via SOM neurons cannot disclose whether the E neurons operate in the ISN or non-ISN regime if SOM neurons also suppress the activity of E neurons. Rather, one should observe a paradoxical response of the total inhibitory current (from PV and SOM) onto E neurons to establish that the network is in the ISN regime (*Litwin-Kumar et al., 2016*).

## Quantifying network stability

The Jacobian matrix of the system is given by

$$\mathbb{J} = \mathbf{W} - \mathbf{B}^{-1} \tag{18}$$

which can be linked to the response matrix since $\mathbf{L} = (\mathbf{B}^{-1} - \mathbf{W})^{-1} = (-\mathbb{J})^{-1}$(*Palmigiano et al., 2023*). The system is stable if the real parts of all three Eigenvalues of the Jacobian are negative. The eigenvalue closest to zero dominates the long term behavior of the system. We quantify stability by measuring the distance of the Eigenvalue with the largest real part $\lambda_{\text{max}}$ to zero (see *Figure 2Biii,Diii*). This stability measure ignores the oscillatory behavior of the system (i.e. the imaginary part of the eigenvalues).

As mentioned in the results section the stability measure can show discontinuities when changing either the rate (*Figure 3iii*) of a population or a synaptic weight (*Figure 7*). This discontinuity follows from either switches of the leading Eigenvalue or changes from non-oscillatory to oscillatory dynamics (*Figure 3iv*).

## Noisy input and numerical measurement of stability and gain

We consider a temporally smoothed input process $\xi_X$ with white noise $\zeta$ (zero mean, standard deviation one): $\tau_\xi\frac{d\xi_X}{dt} = -\xi_X + I_X + \sigma_X\zeta$ for populations $X \in \{E, P\}$ with timescale $\tau_\xi = 50ms$, $\sigma_X = 6$ and fixed mean input $I_X$. To quantify the stability of the network without linearization, we assume that a network is more stable if the mean and variance of excitatory rates are low. To quantify network gain, we freeze the white noise process $\zeta$ for the case of with and without stimulus presentation

and calculate the difference of E rates at each time point, leading to a distribution of network gains (*Figure 6Cii,Fii*). Total simulation time is 1000 s.

## Modulation of tuned populations

We separate the input to each population into two components, a background and a tuned input $\mathbf{I} = \mathbf{I}^{\text{back}} + \mathbf{I}^{\text{stim}}$. We assume that the feedforward stimulus input is tuned with a Gaussian profile and that it only targets E and PV neurons:

$$\mathbf{I}^{\text{stim}}(\theta) = w^{ff} e^{-(\theta - \theta^p)^2/\sigma_\theta^2} \begin{pmatrix} 1 \\ 1 \\ 0 \end{pmatrix}, \tag{19}$$

with $w^{ff} = 2$, the preferred angle $\theta^p = 90°$ and $\sigma_\theta = 20$. For simplicity, we assume that E and PV receive the exact same input tuning. The background input is $\mathbf{I}^{\text{back}} = (1, 1, 1.5)^T$. In *Figure 8* we compare five different circuits, where the E–PV weight strength is fixed and we change the connections to and from SOM.

To quantify if changes in tuning curves are additive/subtractive or mulitplicative/divisive, we use the same measure as in experimental studies (*Phillips and Hasenstaub, 2016*; *Arandia-Romero et al., 2016*). We fit a line to the rates before versus after SOM modulation. The tuning curve undergoes a multiplicative change if the slope is >1, and a divisive change if the slope is <1. If the intersect with the y-axis is >0, the tuning curve change has an additive component and if the intersect is <0 the change has a subtractive component (*Figure 8A*; bottom).

## Code

Code to replicate simulation and theory results is freely available at https://github.com/brain-math/stability-gain-with-multiple-INs (copy archived at *Doiron lab, 2025*).

## Acknowledgements

We thank Xinruo Yang, Fereshteh Lagzi, and Gregory Handy for useful comments on the manuscript. Funding was provided by the National Institutes of Health Grants 1U19NS107613 (BD), CRCNS R01DC015139 (AMO, BD), and R01EB026953 (BD), the Vannevar Bush Faculty Fellowship ONR-N00014-18-1-2002 (BD, AMO), an award from the Simons Foundation Collaboration on the Global Brain 542967 (BD), and a Human Frontier Science Program Postdoctoral Fellowship LT0005/2024 L (CM).

## Additional information

### Funding

| Funder | Grant reference number | Author |
|---|---|---|
| Human Frontier Science Program | LT0005/2024-L | Christoph Miehl |
| NIH Blueprint for Neuroscience Research | 1U19NS107613 | Brent Doiron |
| NIH Blueprint for Neuroscience Research | R01DC015139 | Anne-Marie Michelle Oswald |
| Office of Naval Research | N00014-18-1-2002 | Brent Doiron |
| Simons Foundation | 542967 | Brent Doiron |
| National Institutes of Health | R01NS133598 | Brent Doiron |

The funders had no role in study design, data collection and interpretation, or the decision to submit the work for publication.

## Author contributions

Hannah Bos, Christoph Miehl, Conceptualization, Software, Formal analysis, Visualization, Writing - original draft, Writing - review and editing; Anne-Marie Michelle Oswald, Conceptualization, Supervision, Funding acquisition, Writing - review and editing; Brent Doiron, Conceptualization, Supervision, Funding acquisition, Writing - original draft, Writing - review and editing

## Author ORCIDs

Christoph Miehl (ID) https://orcid.org/0000-0001-9094-2760
Anne-Marie Michelle Oswald (ID) https://orcid.org/0000-0003-1529-1499
Brent Doiron (ID) https://orcid.org/0000-0002-6916-5511

Reviewer #1 (Public review): https://doi.org/10.7554/eLife.99808.4.sa1
Reviewer #2 (Public review): https://doi.org/10.7554/eLife.99808.4.sa2
Reviewer #3 (Public review): https://doi.org/10.7554/eLife.99808.4.sa3
Author response https://doi.org/10.7554/eLife.99808.4.sa4

## Additional files

### Supplementary files

MDAR checklist

### Data availability

All code can be found on GitHub in the repository at https://github.com/brain-math/stability-gain-with-multiple-INs (copy archived at *Doiron lab, 2025*).

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
