## [Editor Report · eLife Assessment]

This paper explores how diverse forms of inhibition impact firing rates in models for cortical circuits. In particular, the paper studies how the network operating point affects the balance of direct inhibition from SOM inhibitory neurons to pyramidal cells, and disinhibition from SOM inhibitory input to PV inhibitory neurons. This is an **important** issue as these two inhibitory pathways have largely been studied in isolation. A combination of analytical calculations and direct numerical simulations provides **convincing** evidence that the interplay of these inhibitory circuits can separately control network gain and stability.

---

## [Referee Report · Reviewer #1 (Public review)]

Summary:

This paper explores how diverse forms of inhibition impact firing rates in models for cortical circuits. In particular, the paper studies how the network operating point affects the balance of direct inhibition from SOM inhibitory neurons to pyramidal cells, and disinhibition from SOM inhibitory input to PV inhibitory neurons. This is an important issue as these two inhibitory pathways have largely been studies in isolation. A combination of analytical calculations and direct numerical simulations provide convincing evidence that the interplay of these inhibitory circuits can separately control network gain and stability.

Strengths

The paper has improved in revision, and the intuitive summary statements added to the end of each results section are quite helpful. The addition of numerical simulations to extend the conclusions beyond the linear range of network behavior are also quite helpful.

Weaknesses

None

---

## [Referee Report · Reviewer #2 (Public review)]

Summary:

Bos and colleagues address the important question of how two major inhibitory interneuron classes in the neocortex differentially affect cortical dynamics. They address this question by studying Wilson-Cowan-type mathematical models. Using a linearized fixed point approach, and subsequent simulations of neural circuits operating in the dynamic stochastically-driven regime, they provide compelling evidence that the existence of multiple interneuron classes can explain the counterintuitive finding that inhibitory modulation can increase the gain of the excitatory cell population while also increasing the stability of the circuit's state to minor perturbations. This effect depends on the connection strengths within their circuit model, providing important guidance as to when and why it arises.

Overall, I find this study to have substantial merit. The authors have also done a commendable job of revising the paper in light of the critiques raised by myself and the other reviewers.

Strengths:

(1) The thorough investigation of how changes in the connectivity structure affect the gain-stability relationship is a major strength of this work. It provides an opportunity to understand when and why gain and stability will or will not both increase together. It also provides a nice bridge to the experimental literature, where different gain-stability relationships are reported from different studies.

(2) The simplified and abstracted mathematical model has the benefit of facilitating our understanding of this puzzling phenomenon. It is not easy to find the right balance between biologically-detailed models vs simple but mathematically tractable ones, and I think the authors struck an excellent balance in this study.

(3) While the fixed-point analysis has potentially substantial limitations for understanding cortical computations away from the steady-state, the authors used simulations to verify that their main findings hold in the stochastically-driven regime that more closely reflects the dynamics observed in in vivo neuroscience experiments.

Weaknesses:

(1) As the authors note in their Discussion, it would be worthwhile to study this effect in chaotic and/or oscillatory regimes, in addition to the ones they included here. I agree with their assessment that those investigations should be left for a future study.

(2) The analysis is limited to paths within this simple E,PV,SOM circuit. This misses more extended paths (like thalamocortical loops) that involve interactions between multiple brain areas. Including those paths in the expansion in Eqs. 11-14 (Fig. 1C) may be an important direction for future work.

---

## [Referee Report · Reviewer #3 (Public review)]

Summary:

Bos et al study a computational model of cortical circuits with excitatory (E) and two subtypes of inhibition - parvalbumin (PV) and somatostatin (SOM) expressing interneurons. They perform stability and gain analysis of simplified models with nonlinear transfer functions when SOM neurons are perturbed. Their analysis suggests that in a specific setup of connectivity, instability and gain can be untangled, such that SOM modulation leads to both increase in stability and gain, in contrast to the typical direction in neuronal networks where increased gain results in decreased stability.

Strengths:

- Analysis of the canonical circuit in response to SOM perturbations. Through numerical simulations and mathematical analysis, the authors have provided a rather comprehensive picture of how SOM modulation may affect response changes.

- Shedding light on two opposing circuit motifs involved in the canonical E-PV-SOM circuitry - namely, direct inhibition (SOM -> E) vs disinhibition (SOM -> PV -> E). These two pathways can lead to opposing effects, and it is often difficult to predict which one results from modulating SOM neurons. In simplified circuits, the authors show how these two motifs can emerge and depend on parameters like connection weights.

- Suggesting potentially interesting consequences for cortical computation. The authors suggest that certain regimes of connectivity may lead to untangling of stability and gain, such that increases in network gain are not compromised by decreasing stability. They also link SOM modulation in different connectivity regimes to versatile computations in visual processing in simple models.

Weaknesses:

- Computationally, the analysis is solid, but it's very similar to previous studies (del Molino et al, 2017). Many studies in the past few years have done the perturbation analysis of a similar circuitry with or without nonlinear transfer functions (some of them listed in the references). This study applies the same framework to SOM perturbations, which is a useful computational analysis, in view of the complexity of the high-dimensional parameter space.

- A general weakness of the paper is a lack of direct comparison to biological parameters or experiments. How different experiments can be reconciled by the results obtained here, and what new circuit mechanisms can be revealed? In its current form, the paper reads as a general suggestion that different combinations of gain modulation and stability can be achieved in a circuit model equipped with many parameters (12 parameters). This is potentially interesting but not surprising, given the high dimensional space of possible dynamical properties. A more interesting result would have been to relate this to biology, by providing reasoning why it might be relevant to certain circuits (and not others), or to provide some predictions or postdictions, which are currently not very strong in the manuscript.

- Tuning curves are simulated for an individual orientation (same for all neurons), not considering the heterogeneity of neuronal networks with multiple orientation selectivity (and other visual features) - making the model too simplistic.

---

## [Author Response]

The following is the authors’ response to the previous reviews

**Reviewer #1 (Public Review):**
Summary:This paper explores how diverse forms of inhibition impact firing rates in models for cortical circuits. In particular, the paper studies how the network operating point affects the balance of direct inhibition from SOM inhibitory neurons to pyramidal cells, and disinhibition from SOM inhibitory input to PV inhibitory neurons. This is an important issue as these two inhibitory pathways have largely been studies in isolation. Support for the main conclusions is generally solid, but could be strengthened by additional analyses.StrengthsThe paper has improved in revision, and the new intuitive summary statements added to the end of each results section are quite helpful. WeaknessesThe concern about whether the results hold outside of the range in which neural responses are linear remains. This is particularly true given the discontinuity observed in the stability measure. I appreciate the concern (provided in the response to the first round of reviews) that studying nonlinear networks requires a lot of work. A more limited undertaking would be to test the behavior of a spiking network at a few key points identified by your linearization approach. Such tests could use relatively simple (and perhaps imperfect) measures of gain and stability. This could substantially enhance the paper, regardless of the outcome.

We appreciate the reviewer’s concern and in our resubmission we explore if networks dynamics that operate outside of the case where linearization is possible would continue to show our main result on the (dis)entanglement of stability and gain; the short answer is yes. To this end we have added a new section and Figure to our main text.

“Gain and stability in stochastically forced E – PV – SOM circuits

To confirm that our results do not depend on our approach of a linearization around a fixed point, we numerically simulate similar networks as shown above (Figure 2) in which the E and PV population receive slow varying, large amplitude noise (Figure 6A). This leads to noisy rate dynamics sampling a large subspace of the full firing rate grid (*rE,rP*) and thus any linearization would fail to describe the network response. In this stochastically forced network we explore how adding an SOM modulation or a stimulus affects this subspace (Figure 6B). To quantify stability without linearization, we assume that a network is more stable the lower the mean and variance of E rates. This is because very stable networks can better quench input fluctuations [Kanashiro et al., 2017; Hennequin et al., 2018]. To quantify gain, we calculate the change in E rates when adding the stimulus, yet having identical noise realizations for stimulated and non-stimulated networks (Methods).

For the disinhibitory network without feedback a positive SOM modulation decreases stability due to increases of the mean and variance of E rates (Figure 6Ci) while the network gain increases (Figure 6Cii). As seen before (Figure 2A,B), stability and gain change in opposite directions in a disinhibitory circuit without feedback. Adding feedback PV → SOM and applying a negative SOM modulation increases both, stability and gain and therefore disentangles the inverse relation also in a noisy circuit (Figure 6D-F). This gives numerical support that our results do not depend on the assumption of linearization.

“Methods: Noisy input and numerical measurement of stability and gain

We consider a temporally smoothed input process *ξX* with white noise *ζ* (zero mean, standard deviation one): τξdξXdt=−ξX+IX+σXζ for populations *X* ∈{*E,P*} with timescale *τξ* = 50ms, *σX* = 6 and fixed mean input *IX*. To quantify the stability of the network without linearization, we assume that a network is more stable if the mean and variance of excitatory rates are low. To quantify network gain, we freeze the white noise process *ζ* for the case of with and without stimulus presentation and calculate the difference of E rates at each time point, leading to a distribution of network gains (Figure 6Cii,Fii). Total simulation time is 1000 seconds.”

We decided against using a spiking network because sufficiently asynchronous spiking network dynamics can still obey a linearized mean field theory (if the fluctuations in population firing rates are small). In our new analysis the firing rate deviations from the time averaged firing rate are sizable, making a linearization ineffective.

In summary, based on our additional analysis of recurrent circuits with noisy inputs we conclude that our results also hold in fluctuating networks, without the need of assuming realization aroud a stable fixed point.

**Reviewer #2 (Public Review):**
Summary:Bos and colleagues address the important question of how two major inhibitory interneuron classes in the neocortex differentially affect cortical dynamics. They address this question by studying Wilson-Cowan-type mathematical models. Using a linearized fixed point approach, they provide convincing evidence that the existence of multiple interneuron classes can explain the counterintuitive finding that inhibitory modulation can increase the gain of the excitatory cell population while also increasing the stability of the circuit’s state to minor perturbations. This effect depends on the connection strengths within their circuit model, providing valuable guidance as to when and why it arises.Overall, I find this study to have substantial merit. I have some suggestions on how to improve the clarity and completeness of the paper.Strengths:(1) The thorough investigation of how changes in the connectivity structure affect the gain-stability relationship is a major strength of this work. It provides an opportunity to understand when and why gain and stability will or will not both increase together. It also provides a nice bridge to the experimental literature, where different gain-stability relationships are reported from different studies.(2) The simplified and abstracted mathematical model has the benefit of facilitating our understanding of this puzzling phenomenon. (I have some suggestions for how the authors could push this understanding further.) It is not easy to find the right balance between biologically-detailed models vs simple but mathematically tractable ones, and I think the authors struck an excellent balance in this study.

We thank the reviewer for their support of our work.

Weaknesses:(1) The fixed-point analysis has potentially substantial limitations for understanding cortical computations away from the steady-state. I think the authors should have emphasized this limitation more strongly and possibly included some additional analyses to show that their conclusions extend to the chaotic dynamical regimes in which cortical circuits often live.

In the response to reviewer 1 we have included model analyses that addresses the limitations of linearization. Rather than use a chaotic model, which would require significant effort, we opted for a stochastically forced network, where the sizable fluctuations in rate dynamics preclude linearization.

(2) The authors could have discussed – even somewhat speculatively – how VIP interneurons fit into this picture. Their absence from this modelling framework stands out as a missed opportunity.

We agree that including VIP neurons into the framework would be an obvious and potentially interesting next step. At this point we only include them as potential modulators of SOM neurons. Modeling their dynamics without them receiving inputs from E, PV, or SOM neurons would be uninteresting. However, including them properly into the circuit would be outside the scope of the paper.

(3) The analysis is limited to paths within this simple E, PV, SOM circuit. This misses more extended paths (like thalamocortical loops) that involve interactions between multiple brain areas. Including those paths in the expansion in Eqs. 11-14 (Fig. 1C) may be an important consideration.

We agree that our pathway expansion can be used to study more than just the E – PV – SOM circuit. However, properly investigating full thalamocortcial loops should be done in a subsequent study.

Comments on revisions:I think the authors have done a reasonable job of responding to my critiques, and the paper is in pretty good shape. (Also, thanks for correctly inferring that I meant VIP interneurons when I had written SST in my review! I have updated the public review accordingly.)I still think this line of research would benefit substantially from considering dynamic regimes including chaotic ones. I strongly encourage the authors to consider such an extension in future work.

Please see our response above to Reviewer 1.

**Reviewer #3 (Public Review):**
Summary:Bos et al study a computational model of cortical circuits with excitatory (E) and two subtypes of inhibition parvalbumin (PV) and somatostatin (SOM) expressing interneurons. They perform stability and gain analysis of simplified models with nonlinear transfer functions when SOM neurons are perturbed. Their analysis suggests that in a specific setup of connectivity, instability and gain can be untangled, such that SOM modulation leads to both increases in stability and gain, in contrast to the typical direction in neuronal networks where increased gain results in decreased stability.Strengths:- Analysis of the canonical circuit in response to SOM perturbations. Through numerical simulations and mathematical analysis, the authors have provided a rather comprehensive picture of how SOM modulation may affect response changes.- Shedding light on two opposing circuit motifs involved in the canonical E-PV-SOM circuitry - namely, direct inhibition (SOM -> E) vs disinhibition (SOM -> PV -> E). These two pathways can lead to opposing effects, and it is often difficult to predict which one results from modulating SOM neurons. In simplified circuits, the authors show how these two motifs can emerge and depend on parameters like connection weights.- Suggesting potentially interesting consequences for cortical computation. The authors suggest that certain regimes of connectivity may lead to untangling of stability and gain, such that increases in network gain are not compromised by decreasing stability. They also link SOM modulation in different connectivity regimes to versatile computations in visual processing in simple models.

We thank the reviewer for their support of our work.

WeaknessesComputationally, the analysis is solid, but it’s very similar to previous studies (del Molino et al, 2017). Many studies in the past few years have done the perturbation analysis of a similar circuitry with or without nonlinear transfer functions (some of them listed in the references). This study applies the same framework to SOM perturbations, which is a useful computational analysis, in view of the complexity of the high-dimensional parameter space.Link to biology: the most interesting result of the paper with regard to biology is the suggestion of a regime in which gain and stability can be modulated in an unconventional way - however, it is difficult to link the results to biological networks:- A general weakness of the paper is a lack of direct comparison to biological parameters or experiments. How different experiments can be reconciled by the results obtained here, and what new circuit mechanisms can be revealed? In its current form, the paper reads as a general suggestion that different combinations of gain modulation and stability can be achieved in a circuit model equipped with many parameters (12 parameters). This is potentially interesting but not surprising, given the high dimensional space of possible dynamical properties. A more interesting result would have been to relate this to biology, by providing reasoning why it might be relevant to certain circuits (and not others), or to provide some predictions or postdictions, which are currently missing in the manuscript.- For instance, a nice motivation for the paper at the beginning of the Results section is the different results of SOM modulation in different experiments - especially between L23 (inhibition) and L4 (disinhibition). But no further explanation is provided for why such a difference should exist, in view of their results and the insights obtained from their suggested circuit mechanisms. How the parameters identified for the two regimes correspond to different properties of different layers?

Please see our answer to the previous round of revision.

- One of the key assumptions of the model is nonlinear transfer functions for all neuron types. In terms of modelling and computational analysis, a thorough analysis of how and when this is necessary is missing (an analysis similar to what has been attempted in Figure 6 for synaptic weights, but for cellular gains). A discussion of this, along with the former analysis to know which nonlinearities would be necessary for the results, is needed, but currently missing from the study. The nonlinearity is assumed for all subtypes because it seems to be needed to obtain the results, but it’s not clear how the model would behave in the presence or absence of them, and whether they are relevant to biological networks with inhibitory transfer functions.

Please see our answer to the previous round of revision.

- Tuning curves are simulated for an individual orientation (same for all), not considering the heterogeneity of neuronal networks with multiple orientation selectivity (and other visual features) - making the model too simplistic.

Please see our answer to the previous round of revision.

**Reviewer #1 (Recommendations For The Authors):**
Introduction, first paragraph, last sentence: suggest ”sense,” -> ”sense” (no comma)Introduction, second paragraph, first sentence: suggest ”is been” -> ”has been”Introduction, very end of next to last paragraph: clarify ”modulate the circuit”Figure 1 legend: can you make the ”Change ...” in the legend for 1D clearer - e.g. ”strenghen SOM → E connections and eliminate SOM → P connections”.Paragraph immediately below Figure 1: In sentence starting ”Specifically ...” can you relate the cases described here back to the equation in Figure 1C?Sentence right below equation 2: This sentence does not separate the network gain from the cellular gain as clearly as it could.Page 7, second full paragraph: sentence starting ”Therefore, with ...” could be split into two or otherwise made clearer.Sentence starting ”Furthermore” right below Figure 5 has an extra comma

We thank the reviewer for their additional comments, we made the respective changes in the manuscript.

**Reviewer #3 (Recommendations For The Authors):**
There is a long part in the reply letter discussing the link to biology - but the revised manuscript doesn’t seem to reflect that.The information in the reply letter discussing the link to biology has been added at multiple points in the discussion. In the section ‘decision of labor between PV and SOM neurons’ we mention Ferguson and Carding 2020, in the section ‘impact of SOM neuron modulation on tuning curves’ we discuss Phillups and Hasenstaub 2016, and in the section ‘limitations and future directions’ we mention Tobin et al., 2023.The writing can be improved - for example, see below instances:P. 7: Intuitively, the inverse relationship follows for inhibitory and disinhibitory pathways (and their mixture) because the firing rate grid (heatmap) does not depend on how the SOM neurons inhibit the E - PV circuit.P.8: We first remark that by adding feedback E connections onto SOM neurons, changes in SOM rates can now affect the underlying heatmaps in the (rE, rP) grid.Not clear how ”rates can affect the heatmaps”. It’s too colloquial and not scientifically rigorous or sound.

We added further explanations at the respective places in the manuscript to improve the writing.